# IMAGE-POSER: REFLECTIVE RL FOR MULTI-EXPERT IMAGE GENERATION AND EDITING

## ABSTRACT

Recent advances in text-to-image generation have produced strong single-shot models, yet no individual system reliably executes the long, compositional prompts typical of creative workflows. We introduce **Image-POSER**, a *reflective* reinforcement learning framework that (i) **orchestrates** a diverse registry of pretrained text-to-image and image-to-image experts, (ii) **handles long-form prompts end-to-end** through dynamic task decomposition, and (iii) **supervises alignment** at each step via structured feedback from a vision–language model critic. By casting image synthesis and editing as a Markov Decision Process, we learn non-trivial expert pipelines that adaptively combine strengths across models. Experiments show that Image-POSER outperforms baselines, including frontier models, across industry-standard and custom benchmarks in alignment, fidelity, and aesthetics, and is consistently preferred in human evaluations. These results highlight that reinforcement learning can endow AI systems with the capacity to autonomously decompose, reorder, and combine visual models, moving towards general-purpose visual assistants.

## 1 INTRODUCTION

Recent advances in image generation models such as GPT Image 1 and Gemini 2.5 Flash have made it possible to synthesize striking imagery from natural language prompts (OpenAI, 2025a; Gemini Team, 2025). Yet these systems often break down when confronted with **long, compositional instructions** that specify multiple objects, spatial relations, or sequential edits. For example, a marketing designer may request: "*Generate a product mockup with three bottles arranged diagonally on a wooden table, each with distinct labels, then restyle the background to match the brand's colors.*" Today's state-of-the-art models frequently miss such details: they miscount objects, ignore edits, or drift in style. Professionals are forced to manually stitch together a pipeline of specialized tools, iterating by trial and error until the output is acceptable. This workflow is inefficient, brittle, and inaccessible to non-experts who need precise, reliable results.

A key limitation of current systems is the lack of **reflection and correction**. Human creators rarely succeed in one shot; they critique, refine, and retry. However, most models attempt a single forward pass (Podell et al., 2023; Duan et al., 2025a; Chen et al., 2023). When the result is misaligned, the user must intervene. This gap is especially costly in domains where **fidelity and consistency are non-negotiable**, such as product design, advertising, and content localization. Even small deviations, such as incorrect object count, a missing logo, or a style mismatch, render outputs unusable.

We introduce **Image-POSER** (Policy-based Orchestration for Sequential Editing using Reflection), a reinforcement learning (RL) framework that embeds reflection into the orchestration of visual experts. Starting from a prompt (and optionally an input image), Image-POSER proceeds in a loop: (i) a lightweight Deep Q-Network (DQN) agent selects an expert from a heterogeneous pool of text-to-image and image-to-image models; (ii) the chosen expert executes an instruction, producing an updated image; (iii) a vision–language model (VLM) critic evaluates the result, providing dense rewards and structured feedback that update the task set; and (iv) an auxiliary LLM module (Extract Command) isolates the next atomic instruction from the revised set, giving the agent a focused objective for the next step. By alternating between acting, critiquing, planning, and refining, Image-POSER can retry failed subtasks, adaptively re-plan, and compose non-trivial expert pipelines, thereby achieving reliability under complexity: the ability to faithfully execute intricate prompts that single-shot generation approaches fail to solve in one pass.

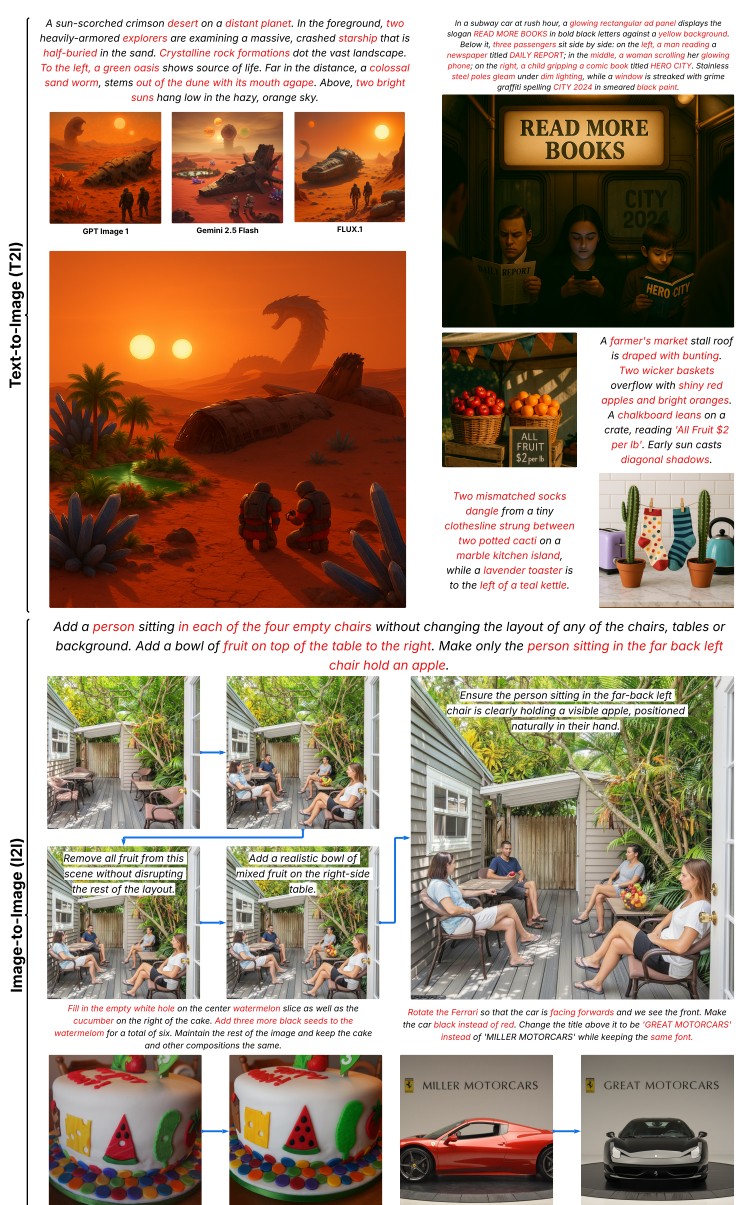

Figure 1: **Select examples for complex long-form compositional prompts.** Top-Left: Text-to-Image (T2I) generations from multiple baselines versus Image-POSER, which successfully integrates all compositional constraints (object counts, spatial relations, style fidelity). Top-Right: additional T2I scenes with fine-grained requirements. Bottom: Image-to-Image (I2I) edits where Image-POSER completes multi-step instructions (adding/removing/counting objects, altering viewpoint, and preserving layout) that single-shot models struggle with. Images cited in Appendix B.

**Contributions** We present Image-POSER, a framework that formulates multi-expert image generation and editing as a sequential decision-making problem, enabling learned orchestration of diverse pretrained models. To support this, we design a reflective RL environment with two complementary modules: one that evaluates intermediate images and updates the set of remaining tasks, and another that extracts a single atomic task to guide the agent's next action. We demonstrate that Image-POSER outperforms both single-model and agentic baselines on compositional benchmarks and user studies, achieving higher fidelity, alignment, and preservation on long-form prompts. By uniting reinforcement learning with image generation and reflective orchestration, Image-POSER moves toward generalist visual agents that can plan, execute, critique, and refine in ways that mirror how skilled artists combine brushes, layers, and filters to realize complex scenes.

## 2 RELATED WORK

Image-POSER is situated at the intersection of three active research directions: (i) orchestration of multiple pretrained visual experts for image generation and editing, (ii) reinforcement learning with model-based feedback, and (iii) reflection and task decomposition for compositional fidelity. While prior works have explored each of these threads separately, our approach unifies them into a single reflective RL framework.

**Image Generation and Editing** A survey of the image generation and editing models used as experts for Image-POSER can be found in Appendix F.

**Orchestration of Multiple Visual Experts** Early work such as *Visual ChatGPT* (Wu et al., 2023) demonstrated how a conversational LLM could call external vision foundation models to perform editing and generation, but relied heavily on user interaction and prompt chaining. *GenArtist* (Wang et al., 2024) extends this paradigm as a successor, replacing interactive dialogue with a multimodal LLM planner that decomposes prompts and executes a fixed schedule of expert calls (e.g., inpainting, style transfer). These approaches highlight the promise of expert orchestration but depend on heuristic planning or prompt engineering. In contrast, Image-POSER learns orchestration policies via reinforcement learning, enabling adaptive expert selection without manual scripting.

**Reinforcement Learning with Feedback for Text–Image Models** RL has also been applied to improve text-image alignment at the single-model level. Reinforcement Learning from AI Feedback (RLAIF) replaces costly human annotation with feedback from powerful LLM or VLM judges. Xu et al. (2023) introduce *ImageReward*, a reward model trained on human preferences that was later used in Reward Feedback Learning (ReFL) to fine-tune diffusion models for better prompt fidelity. These works validate LLM/VLM feedback as a scalable supervisory signal. Image-POSER builds on this principle but shifts the target of learning: instead of tuning one generator, we optimize an orchestration policy over a registry of experts.

**Reflection and Reasoning in Generation** A parallel line of work emphasizes reflection and decomposition as mechanisms to improve compositional fidelity. *GoT-R1* (Duan et al., 2025b), fine-tuned from *Janus-Pro* (Chen et al., 2025), rewards models for performing a "Generation Chain-of-Thought" that decomposes prompts into semantic and spatial components before synthesis. More recently, Venkatesh et al. (2025) propose *CREA*, a collaborative multi-agent framework that mirrors the human creative process through specialized roles such as a Creative Director, Prompt Architect, and Art Critic, coordinating to plan, critique, and refine creative outputs. CREA focuses on creativity-oriented reasoning but uses a fixed backbone, *Flux.1-dev* for generation and *ControlNet* for editing, so orchestration plays a limited role (Labs, 2024; Zhang et al., 2023b). While these systems embed reflection within or across agents, Image-POSER introduces reflection at the orchestration level: a VLM critic evaluates intermediate results, updates residual tasks, and guides expert selection across a heterogeneous pool of experts. This enables retrying failed steps and adaptively refining plans across heterogeneous models, reframing the challenge from improving a single generator to learning how to compose multiple generators and editors.

Taken together, prior work demonstrates the value of orchestration, RL with feedback, and reflection, but each has remained siloed: orchestration has been heuristic, RL has been single-model, and reflection has been intra-model. Image-POSER integrates these threads into a unified reflective RL framework that learns to compose heterogeneous experts into effective pipelines, moving toward general-purpose visual assistants capable of planning, critiquing, and refining complex tasks.

## 3 METHODOLOGY

We formulate multi-step image generation and editing as a sequential decision-making problem. Rather than relying on a fixed pipeline, we treat the generation process as a Markov Decision Process (MDP) where an agent orchestrates a pool of pretrained visual experts under reflective feedback. The agent iteratively invokes text-to-image (T2I) or image-to-image (I2I) experts, and a VLM critic updates the state based on the visual result. The overall process is detailed in Algorithm 1 and illustrated in Figure 2.

## 3.1 PROBLEM FORMULATION

We define the problem as a tuple $\mathcal{M} = \langle \mathcal{S}, \mathcal{A}, \mathcal{T}, \mathcal{R}, \gamma \rangle$. The objective is to learn a policy $\pi : \mathcal{S} \to \mathcal{A}$ that maximizes the cumulative reward by transforming an initial state (blank canvas or input image) into a final image $I_T$ that satisfies a complex prompt $\mathcal{P}$.

**State Space $\mathcal{S}$.** A state $s_t \in \mathcal{S}$ is defined as the tuple:

$$s_t = (\mathcal{P}, I_t, c_t, C_t^{\text{rem}}) \tag{1}$$

where $\mathcal{P}$ is the original global prompt (invariant), $I_t$ is the image at step $t$, $c_t$ is the current atomic command to be executed, and $C_t^{\text{rem}}$ is the set of remaining commands (where each element carries an attempt counter).

**Action Space $\mathcal{A}$.** The action space consists of a discrete registry of $N$ expert models, $\mathcal{E} = \{e_1, \ldots, e_N\}$, partitioned into T2I and I2I experts. At step $t$, the agent selects an action $a_t$ corresponding to an expert $e_{a_t} \in \mathcal{E}$. To ensure validity, the action space is dynamically masked (e.g., prohibiting I2I experts when $I_t$ is null).

**Transition Function $\mathcal{T}$.** The (stochastic) state transition $s_{t+1} \sim \mathcal{T}(s_t, a_t)$ samples the next state based on three functions that capture the system dynamics:

(i) **Identity:** The global intent prompt remains constant, $\mathcal{P}_{t+1} = \mathcal{P}$.

(ii) **Visual Execution:** The selected expert (stochastically) generates the next image based on the current image and command: $I_{t+1} \sim e_{a_t}(I_t, c_t)$.

(iii) **Reflection:** A VLM critic and LLM command extractor evaluate $I_{t+1}$ against $\mathcal{P}$ and $c_t$ to (stochastically) generate the next command set: $(c_{t+1}, C_{t+1}^{\text{rem}}) \sim f_{\text{reflect}}(I_{t+1}, c_t, C_t^{\text{rem}}, \mathcal{P})$.

This formulation relies on the Markov assumption, which holds because the pretrained experts and VLM/LLM reflectors are stateless (memoryless) functions; their outputs depend solely on the inputs provided at time $t$.

## 3.2 STATE REPRESENTATION FOR POLICY LEARNING

While the theoretical state $s_t$ contains the high-dimensional image $I_t$, our efficient DQN agent operates on a compressed representation. We define the policy input vector $\phi(s_t)$ as:

$$\phi(s_t) = \text{embed}([c_t, C_t^{\text{rem}}]) \tag{2}$$

obtained from a pretrained text encoder. Although $I_t$ is not directly inputted to the Q-network, its influence is captured implicitly via the reflective loop: the visual content of $I_t$ determines the feedback from the VLM, which in turn dictates the next textual state $[c_{t+1}, C_{t+1}^{\text{rem}}]$. Note that the encoder function $\phi$ can be viewed as an observation function that induces a form of partial observability that transforms the orchestration problem into a partially observable Markov decision process. However, this partial observability is not due to inherent constraints of the agent, but rather a design decision to keep the framework simple. The policy of the orchestrator does not require any image processing capacity and therefore can remain lightweight.

## 3.3 REWARD FUNCTION AND REFLECTION

The reward function $\mathcal{R}(s_t, a_t, s_{t+1})$ serves as the critic in our loop. It is computed by querying a VLM with the tuple $(I_t, I_{t+1}, c_t, \mathcal{P})$.

**VLM Critic.** The VLM evaluates the alignment of $I_{t+1}$ with the executed command $c_t$ and the global prompt $\mathcal{P}$, returning a scalar score $r_t^{\text{raw}} \in [0, 10]$ based on a rubric assessing content accuracy, spatial configuration, and style. The optimization reward is normalized and penalized for step count:

$$r_t = \frac{r_t^{\text{raw}}}{10} - 0.05t \tag{3}$$

To ensure robustness, the critic updates $C^{\text{rem}}$: if $c_t$ is unfulfilled, it is returned to the queue with an incremented attempt counter. Commands exceeding three attempts are discarded.

**Extract Command Module.** Following the reward calculation, the explicit planner extracts the next highest-priority command (fewer attempts) from the updated set $C_{t+1}^{\text{rem}}$ to become $c_{t+1}$. This ensures the agent always receives a focused, atomic instruction.

## 3.4 LEARNING PROCESS

We approximate the optimal action-value function $Q^*(s, a)$ using a Deep Q-Network (DQN). The Q-network is a lightweight multi-layer perceptron that maps the textual embedding of $s_t$ to Q-values over the available experts. Training follows an $\epsilon$-greedy exploration strategy, with $\epsilon$ linearly annealed from 1.0 to 0.1 over 50% of training horizon to balance exploration and exploitation. The network is optimized with the standard DQN loss, minimizing the mean squared error between the predicted Q-values and the target:

$$\mathcal{L}(\theta) = \mathbb{E}_{(s,a,r,s') \sim \mathcal{B}} \left[ \left( r + \gamma \max_{a'} Q_{\theta^-}(\phi(s'), a') - Q_\theta(\phi(s), a) \right)^2 \right] \tag{4}$$

where $\mathcal{B}$ is the replay buffer (storing transitions), $\theta^-$ are target network parameters (updated every 100 steps), and $\gamma = 0.99$ is the discount factor. This enables the agent to learn non-trivial orchestration policies, such as when to retry a sub-atomic command versus when to switch to a specialized editor, solely through interaction with the expert registry and the VLM critic.

---

**Algorithm 1** Image-POSER Pipeline for Multi-Step Image Generation & Editing

1: **Input:** Dataset of prompts $\mathcal{D}$, expert registry $\mathcal{E}$, max steps $T_{\max} = 6$
2: Initialize DQN agent with Q-network $Q_\theta$, replay buffer $\mathcal{B}$
3: **for** each training episode **do**
4:      Sample $(\mathcal{P}, I_0) \sim \mathcal{D}$
5:      Initialize $c_0 \leftarrow \mathcal{P}$, $C_0^{\text{rem}} \leftarrow \emptyset$
6:      **for** $t = 0$ to $T_{\max}$ **do**
7:          $\phi_t \leftarrow \texttt{embed}([c_t, C_t^{\text{rem}}])$               ▷ Policy Input Vector (Eq. 2)
8:          $a_t \leftarrow \texttt{dqn.predict}(\phi_t)$               ▷ Masked by expert validity
9:          $e_{a_t} \leftarrow \mathcal{E}[a_t]$
10:         $I_{t+1} \leftarrow e_{a_t}(c_t, I_t)$                   ▷ Visual Execution
11:         $(r_t, \tilde{C}) \leftarrow \texttt{vlm\_critic}(I_{t+1}, c_t, C_t^{\text{rem}}, \mathcal{P})$
12:         **if** $c_t$ is incomplete **and** attempts$(c_t) < 3$ **then**
13:             Re-add $c_t$ to $\tilde{C}$ with incremented attempt counter
14:         $c_{t+1}, C_{t+1}^{\text{rem}} \leftarrow \texttt{extract\_command}(\tilde{C})$      ▷ Reflective Planning
15:         $\phi_{t+1} \leftarrow \texttt{embed}([c_{t+1}, C_{t+1}^{\text{rem}}])$
16:         Store transition $(\phi_t, a_t, r_t, \phi_{t+1}, \text{done})$ in $\mathcal{B}$
17:         Sample batch from $\mathcal{B}$, update $Q_\theta$ via gradient descent
18:         **if** $c_{t+1} = \emptyset$ **and** $C_{t+1}^{\text{rem}} = \emptyset$ **then** $\text{done} \leftarrow True$, **break**

---

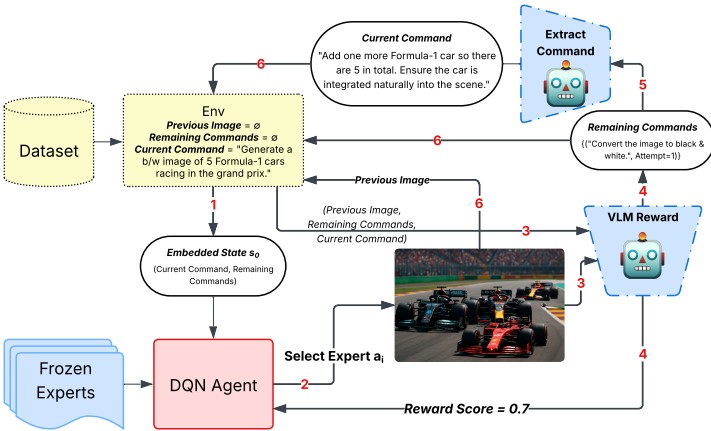

Figure 2: High-level *example flow* of Image-POSER's pipeline for image generation and editing, numbered step by step. Illustrates the RL loop from the environment, to the DQN agent selecting a visual expert, to the VLM outputting a reward and reflecting for future tasks.

## 4 EXPERIMENT AND RESULTS

### 4.1 EXPERIMENTAL SETUP

**Expert Registry** Our framework supports a heterogeneous set of image generation and editing models, unified under a common interface. For text-to-image (T2I) generation, we include *Stable Diffusion XL* (Podell et al., 2023), *PixArt-α* (Chen et al., 2023), *Stable Diffusion 3.5 Large* (Stability AI, 2024), *DALL-E 3* (OpenAI, 2023), *GPT-Image-1* (OpenAI, 2025a), *FLUX.1-dev* (Labs, 2024), and *Gemini 2.5 Flash* (Gemini Team, 2025). For image-to-image (I2I) editing, we support *Instruct-Pix2Pix* (Brooks et al., 2023), *MagicBrush* (Zhang et al., 2023a), *FLUX Kontext* (Labs et al., 2025), *GPT-Image-1* (OpenAI, 2025a), *and Gemini 2.5 Flash* (Gemini Team, 2025).

Each expert was evaluated individually to establish baseline performance. We also compare our system to other existing baselines designed for complex compositional prompts, including *GoT-R1* (Duan et al., 2025b) and *GenArtist* (Wang et al., 2024), which were not part of our registry. All baselines and experts were discussed in Section 2.

**Datasets** For training, we curated a dataset of 450 long-form prompts, combining human-authored and LLM-generated examples. These prompts emphasize multi-object compositions, spatial relations, and stylistic constraints. For evaluation, we considered four datasets: 1) **T2I-CompBench** (Huang et al., 2023): a widely used benchmark for compositional text-to-image generation, 2) **Custom T2I prompts**: 30 long-form generation prompts, designed to probe multi-step reasoning and compositional fidelity, 3) **Custom I2I prompts**: 30 long-form editing prompts with paired input images, targeting object insertion, removal, and style preservation, and 4) **DPG-Bench** (Hu et al., 2024): a large-scale benchmark of 1,065 dense prompts containing many objects, long-range dependencies, and diverse relations.

**Implementation Details** The orchestration agent is a lightweight DQN with a 3-layer MLP Q-network of dimensions $1536 \rightarrow 64 \rightarrow 64 \rightarrow 12$, mapping the text embedding to Q-values over the expert set. For training our DQN, a single NVIDIA T4 GPU was used. For full details regarding our experimental setup, refer to Appendix A and C.

### 4.2 QUANTITATIVE RESULTS

We evaluate Image-POSER across both standardized benchmarks and custom-designed metrics. Three evaluation techniques are used:

1. **T2I-CompBench** (Huang et al., 2023) (Table 1), which measures compositional alignment using pretrained CLIP (Radford et al., 2021) and BLIP (Li et al., 2022) scorers across attribute binding (color, shape, texture), object relationships (spatial, non-spatial), and complex composition.

2. **CLIP/BLIP metrics** (Radford et al., 2021; Li et al., 2022) (Table 6), applied to the outputs of 30 long-form T2I prompts, where BLIP captures binding accuracy and CLIP captures non-spatial relational consistency.

3. **GPT-o3 VLM judgments** (OpenAI, 2025b) (Table 2), used to assess long-form generation and editing tasks. We used GPT-o3 with a modified system prompt that instructed it to output three reward dimensions: alignment, technical (fidelity), and aesthetics (for T2I), or alignment, preservation, and aesthetics (for I2I). This is the same VLM that powers Image-POSER's reward loop, but with a different system prompt, provided in Appendix K.

4. **DPG-Bench** (Hu et al., 2024) (Table 3), which runs each generated image through a large set of VQA queries designed to check whether all entities, attributes, counts, and relationships described in the prompt are correctly visualized.

As shown in Table 1, Image-POSER achieves the strongest scores across all attribute binding categories and outperforms baselines in spatial reasoning. On long-form T2I prompts (Tables 2 and 6), Image-POSER surpasses baselines across both CLIP/BLIP metrics and GPT-o3 VLM judgments, with especially large gains in alignment and technical fidelity. For I2I editing prompts (Table 2), Image-POSER achieves the best overall performance, combining high alignment and aesthetics with competitive preservation scores.

In Table 2, Wilcoxon signed-rank tests (Wilcoxon, 1945) confirm that Image-POSER's improvements over the baselines are statistically significant ($\downarrow$ when $p < 0.05$ and $\Downarrow$ when $p < 0.01$). The only case where a baseline outperformed Image-POSER numerically was in I2I Preservation, but the difference was not statistically significant. These results highlight the robustness of reflective orchestration: by adaptively sequencing expert calls, Image-POSER consistently delivers higher-fidelity outputs across both generation and editing.

As shown in Table 3, Image-POSER achieves the strongest performance on DPG-Bench across nearly all categories (except "Other"). Its ability to preserve object identities and maintain complex relational structure is apparent. Notably, Image-POSER attains the highest "Overall" score, demonstrating that reflective orchestration scales effectively to dense prompts involving many objects and constraints. While GPT-Image-1 performs competitively in the "Other" category, Image-POSER's consistent superiority across the more structured dimensions indicates robustness under long compositional prompts.

Table 1: **T2I-CompBench category scores (higher is better)**. Image-POSER achieves the strongest results across all attribute binding dimensions (color, shape, texture) and outperforms baselines on spatial reasoning. Note * indicates results were sourced from original paper.

| Method | Attribute Binding | | | Object Relationship | | Complex |
|---|---|---|---|---|---|---|
| | Color | Shape | Texture | Spatial | Non-Spatial | |
| *SD XL (Podell et al., 2023) | 0.5879 | 0.4687 | 0.5299 | 0.2133 | 0.3119 | 0.3237 |
| *DALL-E 3 (OpenAI, 2023) | 0.7785 | 0.6205 | 0.7036 | 0.2865 | 0.3003 | 0.3773 |
| *FLUX.1 (Labs, 2024) | 0.7407 | 0.5718 | 0.6922 | 0.2863 | 0.3127 | 0.3703 |
| *GoT-R1-7B (Duan et al., 2025a) | 0.8139 | 0.5549 | 0.7339 | 0.3306 | 0.3169 | **0.3944** |
| *PixArt-$\alpha$ (Chen et al., 2023) | 0.6690 | 0.4927 | 0.6477 | 0.2064 | **0.3197** | 0.3433 |
| GenArtist (Wang et al., 2024) | 0.4775 | 0.4491 | 0.5113 | 0.1587 | 0.2953 | 0.3073 |
| GPT Image 1 (OpenAI, 2025a) | 0.8253 | 0.6145 | 0.7406 | 0.4218 | 0.3158 | 0.3732 |
| Gemini 2.5 Flash (Gemini Team, 2025) | 0.8188 | 0.5572 | 0.7089 | 0.2853 | 0.3069 | 0.3537 |
| *Image-POSER (ours)* | **0.8572** | **0.6218** | **0.7595** | **0.4440** | 0.3195 | 0.3832 |

Table 2: **Quantitative comparison of Image-POSER and baselines on complex long-form T2I/I2I prompts.** Evaluated using a VLM as a judge across 3 key dimensions (higher is better). Arrows indicate statistical significance against Image-POSER under the Wilcoxon signed-rank test. Note that $\downarrow$ ($p < 0.05$) and $\Downarrow$ ($p < 0.01$) indicate Image-POSER performs significantly better than the baseline, while $\uparrow$ and $\Uparrow$ denote the same significance levels when the baseline performs significantly better than Image-POSER.

| Generation Methods | Alignment | Technical | Aesthetic | Average |
|---|---|---|---|---|
| SD 3.5 Large (Stability AI, 2024) | $78.61 \pm 3.21 \Downarrow$ | $93.23 \pm 0.68 \Downarrow$ | $88.67 \pm 1.33 \downarrow$ | $86.84 \pm 1.33 \Downarrow$ |
| SD XL (Podell et al., 2023) | $54.29 \pm 3.48 \Downarrow$ | $91.50 \pm 1.60 \Downarrow$ | $80.00 \pm 2.25 \Downarrow$ | $75.26 \pm 2.21 \Downarrow$ |
| DALL-E 3 (OpenAI, 2023) | $77.43 \pm 3.12 \Downarrow$ | $94.13 \pm 0.85 \Downarrow$ | $90.00 \pm 0.00 \Downarrow$ | $87.19 \pm 1.31 \Downarrow$ |
| FLUX.1 (Labs, 2024) | $81.56 \pm 2.94 \Downarrow$ | $95.87 \pm 0.87 \downarrow$ | $89.60 \pm 0.76 \downarrow$ | $89.01 \pm 1.21 \Downarrow$ |
| GoT-R1-7B (Duan et al., 2025a) | $75.70 \pm 3.14 \Downarrow$ | $91.13 \pm 0.87 \Downarrow$ | $84.67 \pm 2.08 \Downarrow$ | $83.83 \pm 1.44 \Downarrow$ |
| PixArt-$\alpha$ (Chen et al., 2023) | $51.19 \pm 3.51 \Downarrow$ | $92.60 \pm 1.05 \Downarrow$ | $87.20 \pm 1.26 \Downarrow$ | $77.00 \pm 2.33 \Downarrow$ |
| GenArtist (Wang et al., 2024) | $46.37 \pm 3.53 \Downarrow$ | $82.53 \pm 1.61 \Downarrow$ | $61.33 \pm 3.24 \Downarrow$ | $63.41 \pm 2.29 \Downarrow$ |
| GPT Image 1 (OpenAI, 2025a) | $93.80 \pm 1.81 \downarrow$ | $96.93 \pm 0.65$ | $90.40 \pm 0.59$ | $93.71 \pm 0.72 \downarrow$ |
| Gemini 2.5 Flash (Gemini Team, 2025) | $92.12 \pm 1.73 \Downarrow$ | $95.07 \pm 0.92 \Downarrow$ | $89.33 \pm 0.95 \downarrow$ | $92.17 \pm 0.76 \Downarrow$ |
| *Image-POSER (ours)* | **$96.65 \pm 1.01$** | **$97.57 \pm 0.64$** | **$91.33 \pm 0.63$** | **$95.18 \pm 0.53$** |

| Editing Methods | Alignment | Preservation | Aesthetic | Average |
|---|---|---|---|---|
| MagicBrush (Zhang et al., 2023a) | $45.16 \pm 5.14 \Downarrow$ | **$80.93 \pm 3.12$** | $70.83 \pm 3.07 \Downarrow$ | $65.64 \pm 2.74 \Downarrow$ |
| InstructPix2Pix (Brooks et al., 2023) | $33.55 \pm 5.02 \Downarrow$ | $74.50 \pm 5.02 \downarrow$ | $62.27 \pm 3.46 \Downarrow$ | $56.77 \pm 3.18 \Downarrow$ |
| FLUX Kontext (Labs et al., 2025) | $78.98 \pm 2.82 \Downarrow$ | $80.40 \pm 5.08$ | $82.67 \pm 2.03 \Downarrow$ | $80.68 \pm 2.03 \Downarrow$ |
| GPT Image 1 (OpenAI, 2025a) | $91.34 \pm 1.90 \downarrow$ | $70.27 \pm 6.96 \downarrow$ | $89.13 \pm 0.92 \downarrow$ | $83.58 \pm 2.60 \Downarrow$ |
| Gemini 2.5 Flash (Gemini Team, 2025) | $87.79 \pm 2.90 \downarrow$ | $74.43 \pm 6.96$ | $86.43 \pm 1.64 \Downarrow$ | $82.89 \pm 2.62 \downarrow$ |
| *Image-POSER (ours)* | **$94.26 \pm 1.46$** | $80.13 \pm 5.87$ | **$91.07 \pm 1.00$** | **$88.49 \pm 2.12$** |

## 4.3 Qualitative Results

Figures 1 and 3 illustrate qualitative comparisons. They highlight Image-POSER's ability to follow long instructions requiring multiple refinements, such as inserting objects while preserving style or applying sequential edits. In generation tasks, Image-POSER captures both local details (e.g.,

Table 3: **DPG-Bench comparison between Image-POSER and strong baselines.** Higher is better.

| Method | Global | Entity | Attribute | Relation | Other | Overall |
|---|---|---|---|---|---|---|
| GPT-Image-1 (Gemini Team, 2025) | 71.43 | 93.04 | 88.54 | 85.84 | **85.20** | 87.65 |
| Gemini 2.5 Flash (Gemini Team, 2025) | 80.24 | 94.75 | 89.38 | 86.89 | 76.40 | 89.24 |
| *Image-POSER (ours)* | **80.55** | **94.94** | **90.42** | **87.08** | 82.00 | **90.33** |

correct number of players in a sports scene) and global compositional structure (e.g., spatial layout, perspective). In editing tasks, Image-POSER produces accurate, context-aware modifications (e.g., changing object counts, applying style constraints) without degrading unrelated regions.

These examples emphasize the need for multi-step refinement: while strong baselines can produce aesthetic single-shot generations, they often miss compositional constraints. In contrast, Image-POSER incrementally corrects failures, yielding outputs that are both faithful and visually coherent.

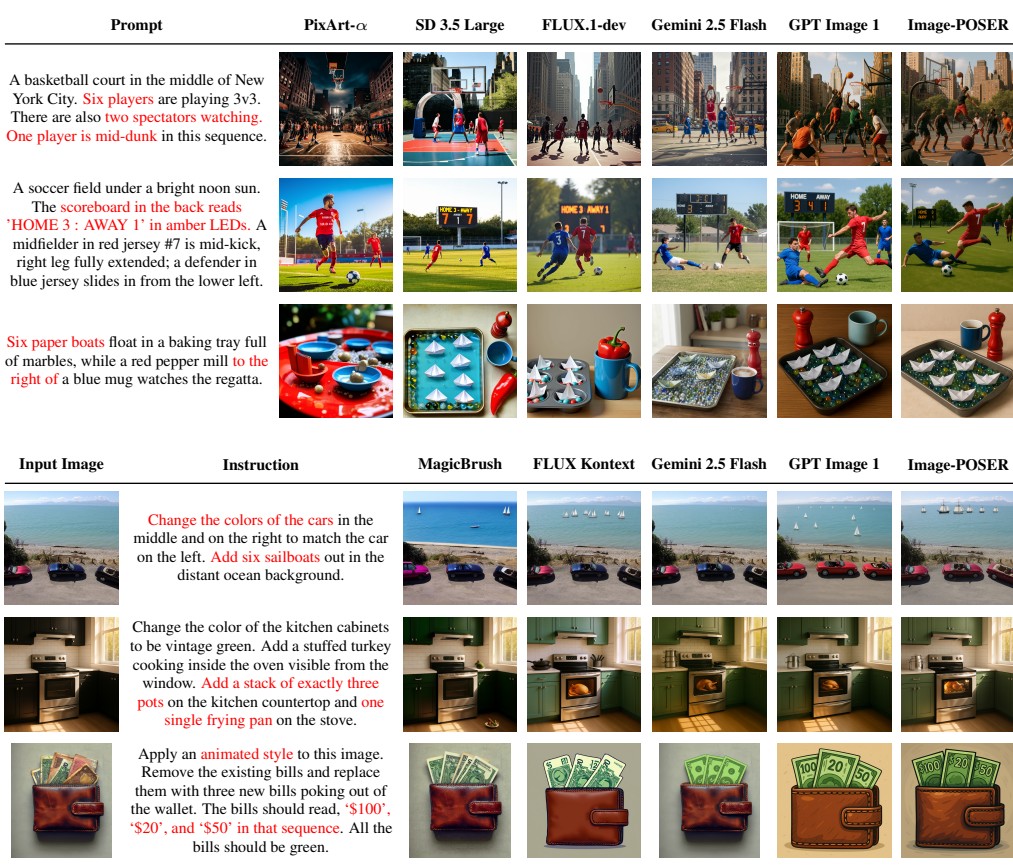

Figure 3: **Qualitative comparison of long-form prompts for generation (top) and editing (bottom).** Baselines often fail on compositional constraints such as object counts, spatial relations, and object addition/removal. Image-POSER produces accurate, context-aware outputs that align with the instructions.

## 4.4 USER STUDY

To complement the automated metrics, we conducted a human preference study. For each setting (generation and editing), 30 prompts were sampled, and participants were shown side-by-side outputs: one from Image-POSER and one from a baseline model (while randomizing the order and hiding which model produced which image). In total, 40 volunteers completed the surveys, and care was taken to ensure that no participant ever saw the same image twice. For more details on this experiment's design, refer to Appendix E.

Table 4: **Win Rates from User Study.** Table shows the average rate at which annotators preferred Image-POSER's outputs over a given baseline (higher is better for Image-POSER). Includes standard errors and arrows indicate statistical significance against Image-POSER ($\downarrow$ $p < 0.05$, $\Downarrow$ $p < 0.01$).

| *Text-to-Image* | | | | | | | | |
|---|---|---|---|---|---|---|---|---|
| **SD 3.5 Large** | **SD XL** | **DALL-E 3** | **FLUX.1** | **GoT-R1-7B** | **PixArt-$\alpha$** | **GenArtist** | **GPT Image 1** | **Gemini 2.5 Flash** |
| $0.75 \pm 0.06 \Downarrow$ | $0.98 \pm 0.02 \Downarrow$ | $0.78 \pm 0.05 \Downarrow$ | $0.80 \pm 0.05 \Downarrow$ | $0.98 \pm 0.02 \Downarrow$ | $0.92 \pm 0.04 \Downarrow$ | $0.95 \pm 0.03 \Downarrow$ | $0.72 \pm 0.04 \Downarrow$ | $0.64 \pm 0.04 \downarrow$ |

| *Image-to-Image* | | | | |
|---|---|---|---|---|
| **MagicBrush** | **InstructPix2Pix** | **FLUX Kontext** | **GPT Image 1** | **Gemini 2.5 Flash** |
| $0.98 \pm 0.02 \Downarrow$ | $0.93 \pm 0.03 \Downarrow$ | $0.83 \pm 0.05 \Downarrow$ | $0.68 \pm 0.04 \Downarrow$ | $0.63 \pm 0.04 \Downarrow$ |

Results are presented in Table 4. In T2I generation, Image-POSER outperformed every baseline, with especially large and statistically significant margins over strong systems such as GPT-Image 1 and Gemini 2.5 Flash, where it more reliably satisfied compositional requirements. In I2I editing, Image-POSER again won against all baselines, including notable gains over MagicBrush and FLUX Kontext, which surpassed Image-POSER on the Preservation dimension in Table 2. Across both tasks, annotators consistently favored Image-POSER for alignment and technical fidelity.

Interestingly, we observed a reversal in relative rankings between Gemini and GPT-Image 1. While GPT-Image 1 generally outperformed Gemini in our automated metrics (Tables 1, 2, 6), Gemini received higher preference in the user study. We attribute this to Gemini's strong rendering fidelity: its images often appear more natural and less overtly "AI-generated," which can make them more appealing in human evaluations even when compositional alignment is imperfect. Importantly, Image-POSER surpassed both models in both studies, indicating that reflective orchestration improves not only automated alignment scores but also human-perceived quality.

## 5 DISCUSSION

### 5.1 REFLECTION ON IMAGE-POSER'S CONTRIBUTIONS

Image-POSER demonstrates that *reflective orchestration* can consistently outperform monolithic generators on complex compositional tasks. By combining retries, dynamic decomposition, and adaptive sequencing of experts, the framework remains robust across diverse prompt types.

A closer analysis clarifies why reflective orchestration is necessary. Figure 6 reports the average reward scores assigned to individual experts, showing that while some models are consistently stronger overall (e.g., GPT Image 1, Gemini 2.5 Flash), others lag behind. Yet Figure 7 reveals that no single model dominates across all task types: one expert may excel in object addition while another performs better in object resizing or background replacement. Task categories were automatically annotated by the *extract command* module, which, alongside extracting $c_t$ and $C_t^{\text{rem}}$, assigned each command to a taxonomy of editing operations (e.g., object addition, removal, resizing). This reveals a central challenge: top-performing models still show significant variability across task types.

The same pattern is evident in the quantitative editing results (Table 2). Image-POSER achieves overall state-of-the-art performance not by uniformly surpassing every expert, but by combining their complementary strengths. For instance, MagicBrush and FLUX Kontext achieve higher Preservation scores than frontier models such as GPT-Image-1 or Gemini 2.5 Flash. However, Image-POSER's preservation performance tracks much closer to MagicBrush and FLUX Kontext than to the frontier models, indicating that it inherits these strengths through orchestration while still maintaining top-tier alignment overall. In practice, reflective orchestration operates as a mixture-of-experts policy: the agent dynamically exploits whichever model is most competent for the current command, rather than depending on a single fixed generator.

Equally important is the framework's *practicality*. Image-POSER does not require retraining any expert models; the only learnable part is a lightweight DQN with a 3-layer MLP. Paired with GPT-o3 for decomposition and reward, this yields a drop-in system that can run on modest compute (single NVIDIA T4 GPU). We see this as a step toward future creative workflows, where the main challenge will be orchestration rather than improving any single model.

Moreover, Image-POSER's modular reward design makes the system *adaptable* to different real-world constraints. Users can easily customize the DQN's reward by adding regularization terms that encode their own time or cost sensitivities, prioritizing faster experts for latency-critical workflows or cheaper experts for budget-constrained settings. This follows the same pattern used in Equation 3, where we incorporated a mild penalty ($-0.05t$) for longer pipelines. This flexibility positions Image-POSER as not merely a fixed policy, but an adaptable orchestrator that aligns with creative and operational preferences.

## 5.2 LIMITATIONS

**Reliance on GPT-○3 as critic/evaluator** Both the *reward* and *extract command* modules use GPT-○3, and GPT-○3 also serves as an automatic evaluator for Table 2. This creates a potential self-preference bias and inherits the usual risks of MLLM hallucination or inconsistency (Panickssery et al., 2024). We chose GPT-○3 pragmatically because it supports reliable multi-image conditioning on $(I_{t-1}, I_t)$ and produced stable judgments in practice. Future work should diversify evaluators (e.g., heterogeneous model ensembles and broader human studies) to reduce any single-model bias.

**Computational cost** Reflective steps introduce non-trivial latency. In our setup, the combined *extract command* and *reward* calls averaged $29.54\,\text{s}$ per step (excluding expert runtime). On average, episodes required 3.37 steps during training and 2.12 steps during inference. This results in some additional end-to-end overhead compared to the average runtime of $62.80\,\text{s}$ for single-shot baselines.

Although baselines can be faster per inference call, they often fail at fully aligning to complex prompts (as shown in Figure 3). In real-world workflows, this failure necessitates human intervention: analyzing the error, re-engineering the prompt, and manually retrying multiple times. The overhead in Image-POSER represents the automation of this "human-in-the-loop" effort. This means that while the wall-clock time is higher, the user's active time is zero, which can be substantially more valuable than agent time. A natural mitigation for this overhead is to distill the critic into a smaller reward model or cache repeated command patterns; we leave a systematic study of such optimizations to future work. The monetary cost of repeated API calls can also be significant, though efficiency gains are possible through shorter episode lengths, batched queries, or adoption of open-source models.

**Evaluation dimensions** Our quantitative metrics emphasize alignment, technical fidelity, preservation (for editing), and aesthetics. We did not directly evaluate creativity or diversity, nor broader intent satisfaction beyond compositional alignment. The observed gap between automated metrics and human preferences (e.g., Gemini's strong perceived realism) underscores the value of expanding the evaluation suite in future work.

## 6 CONCLUSION

Image-POSER points toward a future where orchestration is just as important as model scaling. We see clear opportunities to improve the system by incorporating human feedback or ensemble critics, which would help counterbalance the biases of any single evaluator. As we scale to larger expert pools, this approach can evolve into a fully agentic workflow, where specialized agents are interleaved and steered according to user constraints such as speed, budget, and artistic precision.

Practically, this lowers the barrier to entry, allowing non-experts to execute complex edits that previously required professional tools. However, risks remain, as relying too heavily on automated reflection can lower artistic diversity and may be potentially abused [1]. Ultimately, we view Image-POSER as a co-pilot for design and media: a tool built to amplify human creativity, not automate it away.

---

[1]See Appendix G for a deeper discussion on ethical considerations.

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

# APPENDICES

## A   EXPERIMENTAL SETUP DETAILS

This section provides a detailed overview of the training environment, agent hyperparameters, expert models, and software libraries used to ensure the reproducibility of our results. The code for the project can be found in the attached supplementary materials.

### A.1   RL AGENT AND TRAINING

The orchestration agent is a Deep Q-Network (DQN) implemented using the Stable Baselines3 library (Raffin et al., 2021). The Q-network is a 3-layer MLP with a $(1536 \rightarrow 64 \rightarrow 64 \rightarrow 12)$ architecture, mapping the embedded state representation to Q-values for the 12 experts in our registry. For state representations, we used OpenAI's `text-embedding-3-small` encoder (output dimension 1536) to embed the concatenation of the current and remaining commands, $[c_t, C_t^{\text{rem}}]$.

Training was conducted for 1000 steps on a single NVIDIA T4 GPU. Episodes were capped at a maximum of $T_{\max} = 6$ steps. The reward signal from the VLM critic, originally on a $[0, 10]$ scale, was normalized to $[0, 1]$ and augmented with a step penalty of $-0.05 \cdot t$ to encourage the agent to learn efficient policies. Both the *extract command* and *reward* modules were powered by the GPT-`o3` API.

The specific hyperparameters used for the DQN agent during training are detailed in Table 5.

Table 5: DQN Hyperparameters used for training Image-POSER.

| Hyperparameter | Value |
| --- | --- |
| Learning Rate | $5 \times 10^{-4}$ |
| Optimizer | Adam |
| Discount Factor ($\gamma$) | 0.99 |
| Batch Size | 16 |
| Replay Buffer Size | 500 transitions |
| Learning Starts | 50 steps |
| Target Network Update Interval | 50 steps |
| Exploration Strategy | Linear Epsilon Decay |
| Exploration Fraction | 0.5 (of total timesteps) |
| Initial Epsilon ($\epsilon$) | 1.0 |
| Final Epsilon ($\epsilon$) | 0.1 |

### A.2   STEP BUDGET AND RETRY LIMITS

Below we discuss the rationale behind our choice of the maximum step budget ($T_{\max} = 6$) and the command retry limit (3).

**Step Budget ($T_{\max}$).** The step budget $T_{\max} = 6$ serves as a hard ceiling, as episodes are designed to terminate earlier once the VLM critic deems the prompt satisfied. This specific threshold was selected based on three factors:

1. **Dataset Complexity vs. Observed Behavior:** Our training prompts were constructed to contain between 3 to 5 distinct constraints. A budget of 6 provides a safety margin to cover the worst-case complexity while allowing for error correction. As shown in Figure 4, the agent never hits this limit during inference. As mentioned in 5.2, the average episode length during inference was 2.12 steps. A lower $T_{\max}$ (e.g., 3) would have prematurely truncated valid editing sequences, while a significantly higher budget was unnecessary given the convergence profile.

2. **Visual Degradation:** We qualitatively observed that applying more than 5 sequential image-to-image edits using current diffusion-based editors often results in "generational

drift" or pixel degradation, where background coherence and style fidelity deteriorate. Capping the horizon at 6 mitigates this risk.

3. **Computational Constraints:** Since Image-POSER utilizes large foundation models for reflection, extending the horizon incurs linear increases in both latency and monetary cost. $T_{\max} = 6$ represents a practical ceiling that balances performance with resource efficiency.

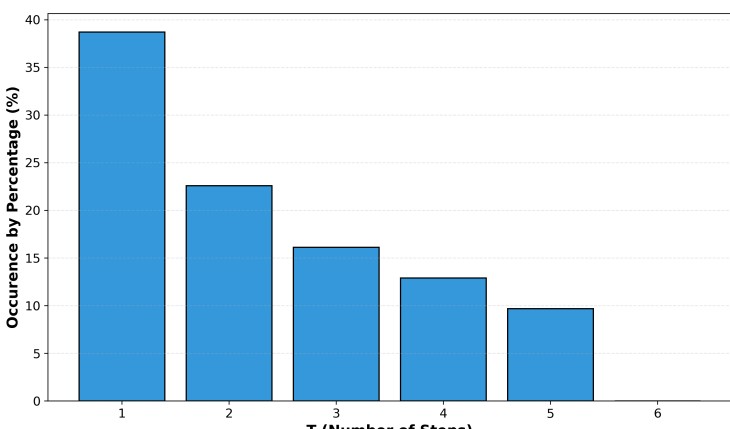

Figure 4: **Distribution of Inference Steps.** A histogram of episode lengths during evaluation. The agent terminates tasks well before reaching the hard limit of $T_{\max} = 6$. This confirms that the budget provides sufficient capacity for complex prompts without inducing unnecessary processing.

**Retry Limit.** We set the retry limit to 3 attempts per atomic command. This threshold distinguishes between *aleatoric uncertainty* (a "bad seed" generation that can be fixed by resampling) and *epistemic limitation* (where the expert fundamentally lacks the ability to perform the specific request).

Qualitatively, this limit acts as a practical safeguard to prevent the agent from becoming trapped in infinite loops due to noisy critic feedback or unresolvable expert failures. Furthermore, the *Extract Command* module is explicitly instructed to prioritize tasks with lower attempt counts, ensuring the agent explores alternative sub-goals before exhausting retries on difficult constraints.

### A.3 EXPERT MODELS AND SOFTWARE

Our expert registry combines open-source models accessed via local inference and proprietary models accessed via APIs.

**Open-Source Models:** We used specific checkpoints hosted on the Hugging Face Hub, listed below:

- `stabilityai/stable-diffusion-3.5-large`
- `stabilityai/stable-diffusion-xl-base-1.0`
- `black-forest-labs/FLUX.1-dev`
- `black-forest-labs/FLUX.1-Kontext-dev`
- `PixArt-alpha/PixArt-XL-2-1024-MS`
- `timbrooks/instruct-pix2pix`
- MagicBrush: `MagicBrush-epoch-52-step-4999.ckpt`

**API-based Models:** The following experts were accessed through their official APIs:

- DALL-E 3
- GPT-Image-1
- Gemini 2.5 Flash

**Software Libraries:** The implementation relies on the Python ecosystem, including `PyTorch`, `Stable Baselines3`, `Transformers`, `diffusers`, `huggingface_hub`, `openai`, and `google.genai`.

## A.4 Reproducibility

For all open-source models and algorithm components, a fixed random seed was used during initialization and training to promote reproducibility. However, we note that results from the closed-source, API-based models may exhibit inherent stochasticity that is beyond our control.

## B Dataset Details

Our experiments rely on a combination of custom-generated prompts for training and evaluation, alongside established academic benchmarks. This approach allows us to train Image-POSER on a diverse set of complex instructions while also measuring its performance against prior work in a standardized manner.

### B.1 Training and Custom Evaluation Prompts

To effectively train and test Image-POSER's ability to handle long-form compositional instructions, we developed a specialized set of prompts for both text-to-image (T2I) generation and image-to-image (I2I) editing.

**Text-to-Image (T2I) Prompts.** We generated a total of 480 long-form T2I prompts: 450 for training the DQN agent and 30 for our custom evaluation set. To ensure diversity in phrasing, complexity, and creative style, we employed a suite of seven powerful Large Language Models: GPT-5 (OpenAI, 2025), Claude 4 Sonnet (Anthropic, 2025), Gemini 2.5 Flash (Gemini Team, 2025), o3 (OpenAI, 2025b), DeepSeek R1 (Ren et al., 2025), Qwen 3 (Alibaba Cloud, 2025), and Llama 4 (Meta AI, 2025).

All models were guided by the unified system prompt shown in Figure 8. This prompt was designed to elicit instructions with multiple objects, specific spatial arrangements, attribute bindings, and stylistic requirements. Below are three representative examples from this dataset.

```
"Snowy street seen through the top windows; inside a cozy coffee shop, an espresso
machine on the left hisses, two ceramic mugs sit centered on a wooden counter, a
dog-eared book on the right lies open."
```

```
"Within a dragon's hoard chamber, treasure-laden chests line the bottom, overflowing
with gold coins; three intricately carved spears lie crossed before an obsidian throne
on the left, while shimmering gemstones spill from a toppled urn in the center, glinting
under the red glow emanating from cracks in the rough cavern walls."
```

```
"Roman forum market at morning, civic and bustling, sunlit travertine arcs throw long
shadows.  Foreground left, a rectangular stall with bronze scales and stacked bread
loaves; foreground right, two terracotta amphorae.  Midground, a marble column fragment
rests near indigo-dyed linen.  Background, repeating arches recede.  Ochre dust lifts
under sandals; awnings billow; polished bronze catches glints; carved Latin letters bite
stone."
```

**Image-to-Image (I2I) Prompts.** The 30 prompts for the I2I evaluation task were manually authored. The 30 input images paired with these prompts were sampled from two high-quality public datasets: **HQ-Edit** (Hui et al., 2024) and **ImgEdit** (Ye et al., 2025). We selected images from these sources for their high resolution and diverse content. Below are a few editing prompt examples:

*Applied to an image of a closed laptop on a table with a pen in front of it:*

```
"Open the laptop to show a space-themed skin covering the entire keyboard surface.
Replace the pen with a small digital alarm clock that reads 03:13 in amber LED light.
The screen of the laptop should be off."
```

*Applied to an image of two sports cars racing in the mountain roads with a scenic background:*

```
"Replace the two red sports cars with two 1800s era wooden horse carriages being pulled
by horses.  One of the horses should be black and the other should be white.  Alter
the paved asphalt to be a dirt road.  Add a wooden sign on the road that says in chalk
'Speed Limit 60'.  Keep the rest of the image as well as the background scenery the
same."
```

## C  ADDITIONAL TRAINING RESULTS

This section provides further insight into the training dynamics of Image-POSER and the performance characteristics of the individual experts within our framework.

Figure 5 shows the DQN agent's learning progress over 1000 training steps. The left plot displays the DQN loss, which decreases rapidly and converges, indicating stable learning. The right plot shows the average cumulative reward, which steadily increases and begins to plateau over the steps. This successful learning trajectory confirms that the agent learned an effective policy for selecting experts.

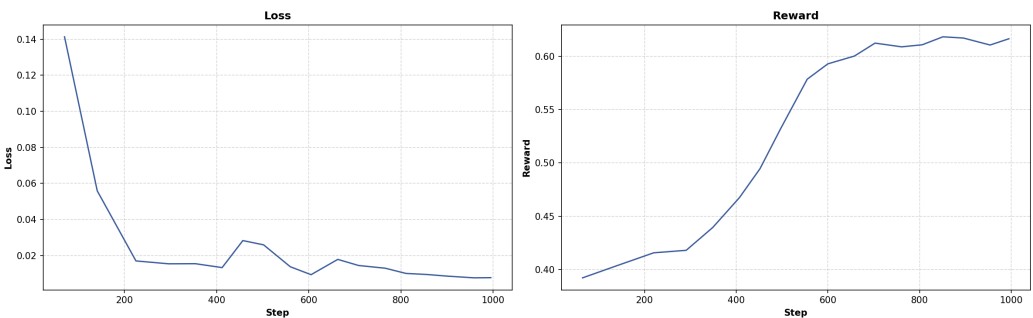

Figure 5: **DQN Training Metrics.** The agent's learning progress over 1000 training steps. The left plot shows the DQN loss, which converges steadily. The right plot shows the cumulative average reward, which increases and plateaus, indicating the agent has learned an effective policy.

Figure 6 provides the average reward scores assigned by the VLM critic to each expert across all tasks during training. This highlights a clear performance hierarchy among the models. Frontier models like GPT Image 1 (T2I) and Gemini 2.5 Flash (T2I) achieve the highest average scores, while more specialized or older models like InstructPix2Pix and MagicBrush score lower on average. This wide distribution of performance across the expert registry underscores the complexity of the orchestration task and justifies the need for an intelligent agent that can learn to select the best model for a given situation, rather than defaulting to a single expert.

Figure 7 further dissects the performance of three powerful I2I experts across nine distinct editing task categories that were tracked during training. The results reveal that no single model is dominant across all tasks. For instance, GPT Image 1 excels at "Add/edit text" (8.25), while FLUX Kontext performs strongly on "Lighting change" (7.67) and "Object resizing" (7.67). Gemini 2.5 Flash shows competitive performance across several areas, such as "Background replacement" (7.67). This data validates the core hypothesis of our work: since expert models have complementary strengths, a dynamic, task-aware orchestration policy that can select the right tool for each sub-task will outperform any single model.

## D  ADDITIONAL EVALUATION RESULTS

This section provides supplementary quantitative results that complement the main VLM-based evaluations presented in Section 4.2.

Table 6 presents the results of this analysis on our custom set of 30 long-form T2I prompts. The BLIP score, which is an average of color, shape, and texture binding accuracy, measures how well models associate attributes with objects. The CLIP score measures non-spatial relational consistency.

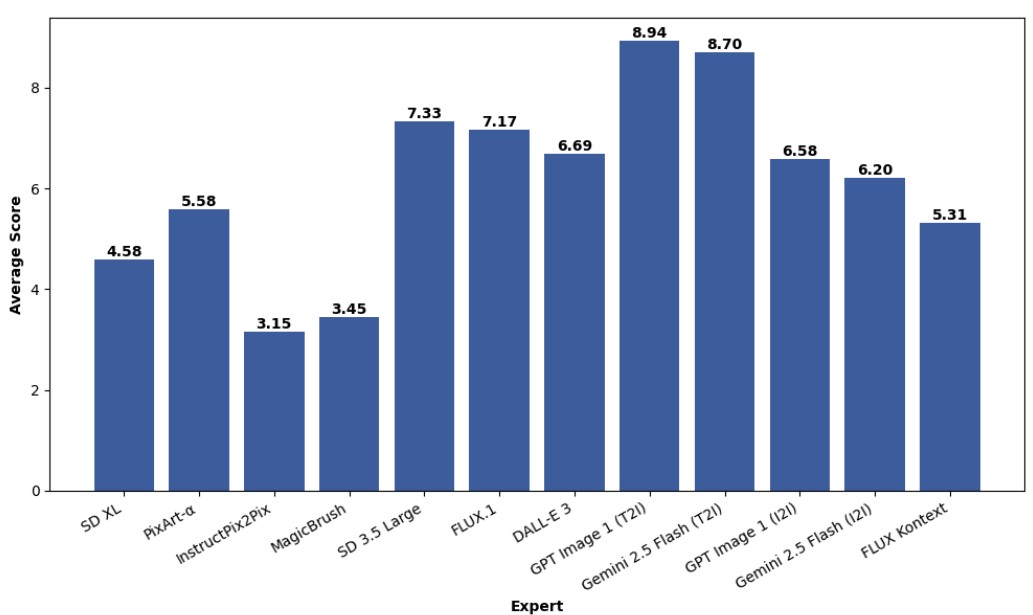

Figure 6: Average reward scores assigned by the VLM critic to each expert during training.

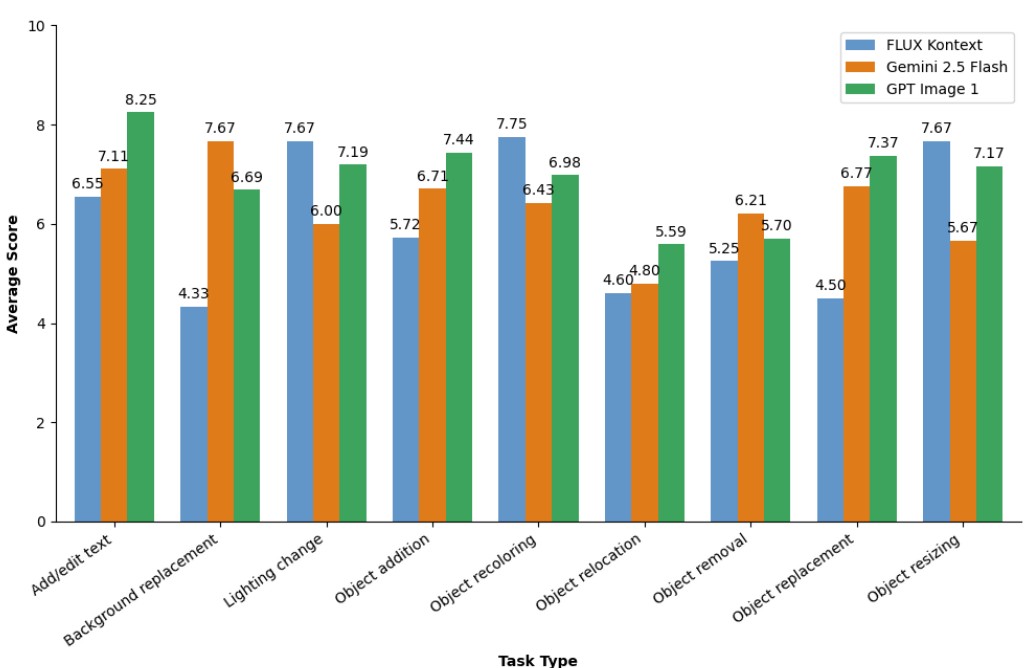

Figure 7: **Task distribution average scores.** Average VLM-assigned scores across editing task types for three strong experts (FLUX Kontext, Gemini 2.5 Flash, GPT Image 1).

Image-POSER achieves the highest scores on both metrics, confirming that its superior performance is not an artifact of the GPT-o3 VLM judge but is also reflected in these established benchmarks. This reinforces the conclusion that our reflective orchestration framework leads to objectively better compositional alignment.

Table 6: **CLIP/BLIP metrics for long-form T2I prompts.** BLIP score is the average of Color, Shape, and Texture binding accuracy. CLIP score measures non-spatial relational consistency. Higher is better for both.

| Method | BLIP | CLIP |
|---|---|---|
| SD 3.5 Large (Stability AI, 2024) | 0.2348 | 0.3494 |
| SD XL (Podell et al., 2023) | 0.0970 | 0.3429 |
| DALL-E 3 (OpenAI, 2023) | 0.1940 | 0.3513 |
| FLUX.1 (Labs, 2024) | 0.1888 | 0.3542 |
| GoT-R1-7B (Duan et al., 2025a) | 0.1991 | 0.3581 |
| PixArt-$\alpha$ (Chen et al., 2023) | 0.0969 | 0.3404 |
| GenArtist (Wang et al., 2024) | 0.0735 | 0.3295 |
| GPT Image 1 (OpenAI, 2025a) | 0.2173 | 0.3630 |
| Gemini 2.5 Flash (Gemini Team, 2025) | 0.2355 | 0.3578 |
| *Image-POSER (ours)* | **0.2419** | **0.3685** |

## E  ADDITIONAL USER STUDY DETAILS

Our study involved 40 participants, all with strong general computer literacy. None had professional experience in computer vision, image/video generation, digital art, or photography. The cohort consisted of 15 AI researchers from broad machine-learning subfields and 25 technically proficient non-experts, all with some experience using AI tools. To focus the evaluation effort where it matters most, we collected five independent ratings for each matchup between Image-POSER and the strongest baselines (GPT-Image-1 and Gemini 2.5 Flash, both for T2I and I2I), and comparisons against weaker baselines received two ratings per prompt.

## F  EXPERTS' SURVEY

**Text-to-Image.** Recent T2I models have steadily advanced in realism, prompt alignment, and speed. Systems such as *Stable Diffusion XL* (Podell et al., 2023), *PixArt-$\alpha$* (Chen et al., 2023), *Stable Diffusion 3.5 Large* (Stability AI, 2024), and *FLUX.1-dev* (Labs, 2024) focus on scaling resolution and improving image quality. In contrast, models like *DALL-E 3* (OpenAI, 2023), *GPT-Image-1* (OpenAI, 2025a), and *Gemini 2.5 Flash* (Gemini Team, 2025) emphasize stronger prompt fidelity and integrated vision–language capabilities, though they typically come with the added cost of API-based access.

**Image-to-Image Editing.** Editing models aim to refine or modify existing images with natural language guidance. Approaches such as *InstructPix2Pix* (Brooks et al., 2023), *MagicBrush* (Zhang et al., 2023a), and *FLUX Kontext* (Labs et al., 2025) enable instruction-based or localized edits. General-purpose systems like *GPT-Image-1* (OpenAI, 2025a) and *Gemini 2.5 Flash* (Gemini Team, 2025) extend their generation pipelines to interactive I2I editing, again at the expense of API usage costs.

## G  ETHICAL CONSIDERATIONS

The development of Image-POSER raises both opportunities and risks. By enabling fine-grained orchestration of powerful generative models, our framework can support beneficial applications in design, accessibility, and education. However, the same capabilities also increase the risk of misuse, such as producing deceptive or harmful content, compounding biases from pretrained experts, or generating outputs that misrepresent individuals or groups. Because Image-POSER relies on reflection and multi-step decision-making, it can potentially reduce oversight compared to interactive, single-shot systems. To mitigate these risks, future deployments should incorporate safeguards such as content moderation filters, watermarking, dataset auditing, and transparency in reporting outputs.

## H ABLATION STUDY: CRITIC AND EXTRACTION MODULES

To assess the robustness of Image-POSER's architecture, we conducted an ablation study on its two core reflective components: the *VLM Critic* (responsible for reward calculation and task updates) and the *Extract Command* module (responsible for decomposing the next atomic instruction).

In our default configuration, both modules utilize GPT-o3. We systematically replaced the *VLM Critic* and the *Extract Command* modules with two alternative state-of-the-art models: **Claude 4.5 Sonnet** (Anthropic, 2025) and **Grok 4 Fast** (xAI, 2025).

Table 7 details the results on our custom T2I benchmark. We observe that Image-POSER maintains state-of-the-art performance regardless of the underlying foundation model used for the reflective components. Whether driven by GPT-o3, Claude, or Grok, the framework consistently outperforms standard single-shot baselines. This confirms that the system's efficacy stems from the orchestration policy and iterative refinement loop, rather than reliance on a single model family.

Table 7: **Robustness Analysis of Reflective Modules.** Comparison of baseline experts against various configurations of Image-POSER. We varied the underlying model for the *Extract Command* module and the *VLM Critic* between GPT-o3 (Default), Claude 4.5 Sonnet, and Grok 4 Fast. The consistently high scores across all configurations show the framework's model-agnostic robustness.

| Method | Alignment | Technical | Aesthetic | Average |
|---|---|---|---|---|
| SD 3.5 Large (Stability AI, 2024) | $78.61 \pm 3.21$ | $93.23 \pm 0.68$ | $88.67 \pm 1.33$ | $86.84 \pm 1.33$ |
| SD XL (Podell et al., 2023) | $54.29 \pm 3.48$ | $91.50 \pm 1.60$ | $80.00 \pm 2.25$ | $75.26 \pm 2.21$ |
| DALL-E 3 (OpenAI, 2023) | $77.43 \pm 3.12$ | $94.13 \pm 0.85$ | $90.00 \pm 0.00$ | $87.19 \pm 1.31$ |
| FLUX.1 (Labs, 2024) | $81.56 \pm 2.94$ | $95.87 \pm 0.87$ | $89.60 \pm 0.76$ | $89.01 \pm 1.21$ |
| GoT-R1-7B (Duan et al., 2025a) | $75.70 \pm 3.14$ | $91.13 \pm 0.87$ | $84.67 \pm 2.08$ | $83.83 \pm 1.44$ |
| PixArt-$\alpha$ (Chen et al., 2023) | $51.19 \pm 3.51$ | $92.60 \pm 1.05$ | $87.20 \pm 1.26$ | $77.00 \pm 2.33$ |
| GenArtist (Wang et al., 2024) | $46.37 \pm 3.53$ | $82.53 \pm 1.61$ | $61.33 \pm 3.24$ | $63.41 \pm 2.29$ |
| GPT Image 1 (OpenAI, 2025a) | $93.80 \pm 1.81$ | $96.93 \pm 0.65$ | $90.40 \pm 0.59$ | $93.71 \pm 0.72$ |
| Gemini 2.5 Flash (Gemini Team, 2025) | $92.12 \pm 1.73$ | $95.07 \pm 0.92$ | $89.33 \pm 0.95$ | $92.17 \pm 0.76$ |
| **Image-POSER (Original: All GPT-o3)** | $96.65 \pm 1.01$ | $97.57 \pm 0.64$ | $91.33 \pm 0.63$ | $95.18 \pm 0.53$ |
| **Image-POSER (Extract: Claude 4.5 Sonnet)** | $96.44 \pm 1.00$ | $97.68 \pm 0.52$ | $90.73 \pm 0.82$ | $94.95 \pm 0.56$ |
| **Image-POSER (Extract: Grok 4 Fast)** | $95.74 \pm 1.39$ | $96.57 \pm 0.66$ | $91.07 \pm 1.11$ | $94.46 \pm 0.68$ |
| **Image-POSER (Critic: Claude 4.5 Sonnet)** | $95.89 \pm 1.21$ | $97.13 \pm 0.50$ | $91.38 \pm 0.64$ | $94.80 \pm 0.55$ |
| **Image-POSER (Critic: Grok 4 Fast)** | $96.32 \pm 1.12$ | $97.70 \pm 0.59$ | $91.73 \pm 0.65$ | $95.25 \pm 0.54$ |

## I ABLATION STUDY: EVALUATOR BIAS

A primary concern in using GPT-o3 as both a training critic and an automatic evaluator is the potential for self-preference bias, where the model may favour outputs that align with its own inductive biases rather than objective quality. To address this, we conducted a robust ablation study by re-evaluating the T2I performance of Image-POSER against the other baselines using two independent, frontier-class Vision-Language Models: **Claude 4.5 Sonnet** (Anthropic, 2025) and **Grok 4 Fast** (xAI, 2025).

We utilized the exact same evaluation system prompts and rubrics as in our main experiments. The results presented in Table 8, demonstrate that the performance of our method is not an artifact of the architecture of the evaluator. Although the absolute scoring distributions shift between judges (with Grok 4 Fast generally assigning higher raw scores), the relative ranking of methods remains consistent. Image-POSER maintains a statistically significant lead in alignment and fidelity across both independent evaluators, confirming that the reported gains reflect genuine improvements in compositional instruction following.

Table 8: **Robustness to Evaluator Choice.** We re-evaluated all methods on our custom T2I benchmark using independent VLM judges (**Claude 4.5 Sonnet** and **Grok 4 Fast**) to rule out self-preference bias from the GPT-o3 critic (Table 2 results included for reference). Image-POSER consistently outperforms baselines regardless of the evaluator used. Arrows indicate statistical significance against Image-POSER ($\downarrow p < 0.05$, $\Downarrow p < 0.01$).

| Method | Alignment | Technical | Aesthetic | Average |
|---|---|---|---|---|
| *Evaluator: GPT-o3 (Original Results)* | | | | |
| SD 3.5 Large (Stability AI, 2024) | $78.61 \pm 3.21 \Downarrow$ | $93.23 \pm 0.68 \Downarrow$ | $88.67 \pm 1.33 \downarrow$ | $86.84 \pm 1.33 \Downarrow$ |
| SD XL (Podell et al., 2023) | $54.29 \pm 3.48 \Downarrow$ | $91.50 \pm 1.60 \Downarrow$ | $80.00 \pm 2.25 \Downarrow$ | $75.26 \pm 2.21 \Downarrow$ |
| DALL-E 3 (OpenAI, 2023) | $77.43 \pm 3.12 \Downarrow$ | $94.13 \pm 0.85 \Downarrow$ | $90.00 \pm 0.00 \downarrow$ | $87.19 \pm 1.31 \Downarrow$ |
| FLUX.1 (Labs, 2024) | $81.56 \pm 2.94 \Downarrow$ | $95.87 \pm 0.87 \downarrow$ | $89.60 \pm 0.76 \downarrow$ | $89.01 \pm 1.21 \Downarrow$ |
| GoT-R1-7B (Duan et al., 2025a) | $75.70 \pm 3.14 \Downarrow$ | $91.13 \pm 0.87 \Downarrow$ | $84.67 \pm 2.08 \Downarrow$ | $83.83 \pm 1.44 \Downarrow$ |
| PixArt-$\alpha$ (Chen et al., 2023) | $51.19 \pm 3.51 \Downarrow$ | $92.60 \pm 1.05 \Downarrow$ | $87.20 \pm 1.26 \Downarrow$ | $77.00 \pm 2.33 \Downarrow$ |
| GenArtist (Wang et al., 2024) | $46.37 \pm 3.53 \Downarrow$ | $82.53 \pm 1.61 \Downarrow$ | $61.33 \pm 3.24 \Downarrow$ | $63.41 \pm 2.29 \Downarrow$ |
| GPT Image 1 (OpenAI, 2025a) | $93.80 \pm 1.81 \downarrow$ | $96.93 \pm 0.65$ | $90.40 \pm 0.59$ | $93.71 \pm 0.72 \Downarrow$ |
| Gemini 2.5 Flash (Gemini Team, 2025) | $92.12 \pm 1.73 \Downarrow$ | $95.07 \pm 0.92 \Downarrow$ | $89.33 \pm 0.95 \downarrow$ | $92.17 \pm 0.76 \Downarrow$ |
| **Image-POSER (ours)** | $\mathbf{96.65 \pm 1.01}$ | $\mathbf{97.57 \pm 0.64}$ | $\mathbf{91.33 \pm 0.63}$ | $\mathbf{95.18 \pm 0.53}$ |
| *Evaluator: Claude 4.5 Sonnet* | | | | |
| SD 3.5 Large (Stability AI, 2024) | $74.69 \pm 3.13 \Downarrow$ | $89.07 \pm 0.89 \Downarrow$ | $84.83 \pm 1.00 \Downarrow$ | $82.86 \pm 1.29 \Downarrow$ |
| SD XL (Podell et al., 2023) | $47.60 \pm 2.81 \Downarrow$ | $82.10 \pm 2.33 \Downarrow$ | $78.77 \pm 1.47 \Downarrow$ | $69.49 \pm 2.10 \Downarrow$ |
| DALL-E 3 (OpenAI, 2023) | $65.74 \pm 3.25 \Downarrow$ | $87.43 \pm 0.95 \Downarrow$ | $84.23 \pm 2.46 \Downarrow$ | $79.14 \pm 1.71 \Downarrow$ |
| FLUX.1 (Labs, 2024) | $77.03 \pm 3.34 \Downarrow$ | $91.03 \pm 0.88 \Downarrow$ | $87.20 \pm 1.06 \downarrow$ | $85.09 \pm 1.34 \Downarrow$ |
| GoT-R1-7B (Duan et al., 2025a) | $77.88 \pm 2.85 \Downarrow$ | $88.77 \pm 1.00 \Downarrow$ | $81.97 \pm 0.97 \Downarrow$ | $82.87 \pm 1.15 \Downarrow$ |
| PixArt-$\alpha$ (Chen et al., 2023) | $44.76 \pm 3.13 \Downarrow$ | $85.50 \pm 1.51 \Downarrow$ | $81.47 \pm 1.21 \Downarrow$ | $70.57 \pm 2.29 \Downarrow$ |
| GenArtist (Wang et al., 2024) | $47.51 \pm 3.42 \Downarrow$ | $73.20 \pm 2.45 \Downarrow$ | $70.70 \pm 1.66 \Downarrow$ | $63.80 \pm 1.93 \Downarrow$ |
| GPT Image 1 (OpenAI, 2025a) | $88.77 \pm 2.44 \downarrow$ | $92.60 \pm 0.67 \Downarrow$ | $89.00 \pm 0.91$ | $90.12 \pm 0.91 \Downarrow$ |
| Gemini 2.5 Flash (Gemini Team, 2025) | $85.46 \pm 2.00 \Downarrow$ | $91.80 \pm 0.65 \downarrow$ | $88.17 \pm 0.96$ | $88.48 \pm 0.81 \Downarrow$ |
| **Image-POSER (ours)** | $\mathbf{92.15 \pm 1.88}$ | $\mathbf{94.10 \pm 0.56}$ | $\mathbf{90.07 \pm 1.17}$ | $\mathbf{92.11 \pm 0.77}$ |
| *Evaluator: Grok 4 Fast* | | | | |
| SD 3.5 Large (Stability AI, 2024) | $87.45 \pm 1.98 \Downarrow$ | $93.33 \pm 0.64 \Downarrow$ | $86.67 \pm 0.84 \downarrow$ | $89.15 \pm 0.81 \Downarrow$ |
| SD XL (Podell et al., 2023) | $62.29 \pm 4.17 \Downarrow$ | $89.40 \pm 1.51 \Downarrow$ | $77.83 \pm 2.63 \Downarrow$ | $76.51 \pm 2.07 \Downarrow$ |
| DALL-E 3 (OpenAI, 2023) | $82.31 \pm 2.88 \Downarrow$ | $94.87 \pm 0.73$ | $87.17 \pm 1.07$ | $88.11 \pm 1.18 \Downarrow$ |
| FLUX.1 (Labs, 2024) | $88.20 \pm 2.64 \Downarrow$ | $94.50 \pm 0.75 \downarrow$ | $86.40 \pm 0.91 \downarrow$ | $89.70 \pm 1.02 \Downarrow$ |
| GoT-R1-7B (Duan et al., 2025a) | $86.00 \pm 2.05 \Downarrow$ | $91.90 \pm 0.79 \Downarrow$ | $81.75 \pm 1.42 \Downarrow$ | $86.55 \pm 0.97 \Downarrow$ |
| PixArt-$\alpha$ (Chen et al., 2023) | $61.85 \pm 3.99 \Downarrow$ | $90.10 \pm 1.83 \Downarrow$ | $79.73 \pm 2.45 \Downarrow$ | $77.23 \pm 2.07 \Downarrow$ |
| GenArtist (Wang et al., 2024) | $58.18 \pm 4.19 \Downarrow$ | $74.57 \pm 3.90 \Downarrow$ | $57.43 \pm 4.22 \Downarrow$ | $63.39 \pm 2.49 \Downarrow$ |
| GPT Image 1 (OpenAI, 2025a) | $94.21 \pm 1.72 \downarrow$ | $95.67 \pm 0.56$ | $88.10 \pm 0.88$ | $92.66 \pm 0.75 \downarrow$ |
| Gemini 2.5 Flash (Gemini Team, 2025) | $93.23 \pm 1.42 \downarrow$ | $94.74 \pm 0.88$ | $87.50 \pm 0.75 \downarrow$ | $91.82 \pm 0.69 \Downarrow$ |
| **Image-POSER (ours)** | $\mathbf{96.90 \pm 1.20}$ | $\mathbf{96.03 \pm 0.60}$ | $\mathbf{89.17 \pm 0.50}$ | $\mathbf{94.03 \pm 0.60}$ |

# J ABLATION STUDY: LEARNED RL POLICY VS. LLM PLANNER

A core design choice in Image-POSER is the use of a Deep Q-Network (DQN) to orchestrate experts, rather than relying on the zero-shot planning capabilities of Large Language Models. To validate this choice, we compared our RL agent against a strong non-learning baseline, referred to here as the **LLM Planner**.

**LLM Planner Setup:** We replaced the DQN agent with **Grok 4 Fast** (xAI, 2025), prompted to act as a heuristic planner. This planner operates in two stages:

1. **Initial Selection:** Based on the prompt $\mathcal{P}$, it selects the most semantically appropriate T2I expert to generate the base image.

2. **Iterative Scheduling:** Upon receiving the image and the VLM critic's feedback, the planner schedules a sequence of edits, assigning each residual task to an I2I expert based on its internal knowledge of model capabilities.

Unlike Image-POSER, the LLM Planner does not update its policy based on success or failure rates; it relies entirely on its pre-trained reasoning.

Table 9 presents the comparison on our custom T2I benchmark. The **LLM Planner** achieves an average score of 89.89, significantly underperforming Image-POSER (95.18). Notably, the LLM Planner also underperforms the strongest single-model baseline (GPT Image 1, 93.71).

This performance gap highlights the limitations of zero-shot planning in multi-expert systems. While the LLM Planner understands the *semantics* of the task, it lacks the empirical experience of expert reliability. It often routes difficult spatial tasks to experts that theoretically should handle them but practically fail (e.g., instructing a specific editor to fix object counts that it historically struggles with). In contrast, the RL agent learns a value function $Q(s, a)$ that implicitly encodes the stochastic success rates of specific experts for specific state configurations, allowing it to avoid brittle pathways that a heuristic planner might choose.

Table 9: **Effectiveness of Learned Orchestration.** We compare Image-POSER against an **LLM Planner** (Grok 4 Fast) that heuristically schedules experts without reinforcement learning. The results show that a learned policy is essential for maximizing performance, as zero-shot planning fails to account for the stochastic reliability of individual experts. Arrows indicate statistical significance against Image-POSER ($\downarrow p < 0.05$, $\Downarrow p < 0.01$).

| Method | Alignment | Technical | Aesthetic | Average |
|---|---|---|---|---|
| GenArtist (Wang et al., 2024) | $46.37 \pm 3.53 \Downarrow$ | $82.53 \pm 1.61 \Downarrow$ | $61.33 \pm 3.24 \Downarrow$ | $63.41 \pm 2.29 \Downarrow$ |
| DALL-E 3 (OpenAI, 2023) | $77.43 \pm 3.12 \Downarrow$ | $94.13 \pm 0.85 \Downarrow$ | $90.00 \pm 0.00 \downarrow$ | $87.19 \pm 1.31 \Downarrow$ |
| FLUX.1 (Labs, 2024) | $81.56 \pm 2.94 \Downarrow$ | $95.87 \pm 0.87 \downarrow$ | $89.60 \pm 0.76 \downarrow$ | $89.01 \pm 1.21 \Downarrow$ |
| GoT-R1-7B (Duan et al., 2025a) | $75.70 \pm 3.14 \Downarrow$ | $91.13 \pm 0.87 \Downarrow$ | $84.67 \pm 2.08 \Downarrow$ | $83.83 \pm 1.44 \Downarrow$ |
| PixArt-$\alpha$ (Chen et al., 2023) | $51.19 \pm 3.51 \Downarrow$ | $92.60 \pm 1.05 \Downarrow$ | $87.20 \pm 1.26 \Downarrow$ | $77.00 \pm 2.33 \Downarrow$ |
| SD XL (Podell et al., 2023) | $54.29 \pm 3.48 \Downarrow$ | $91.50 \pm 1.60 \Downarrow$ | $80.00 \pm 2.25 \Downarrow$ | $75.26 \pm 2.21 \Downarrow$ |
| SD 3.5 Large (Stability AI, 2024) | $78.61 \pm 3.21 \Downarrow$ | $93.23 \pm 0.68 \Downarrow$ | $88.67 \pm 1.33 \downarrow$ | $86.84 \pm 1.33 \Downarrow$ |
| Gemini 2.5 Flash (Gemini Team, 2025) | $92.12 \pm 1.73 \Downarrow$ | $95.07 \pm 0.92 \Downarrow$ | $89.33 \pm 0.95 \downarrow$ | $92.17 \pm 0.76 \Downarrow$ |
| GPT Image 1 (OpenAI, 2025a) | $93.80 \pm 1.81 \downarrow$ | $96.93 \pm 0.65$ | $90.40 \pm 0.59$ | $93.71 \pm 0.72 \Downarrow$ |
| **LLM Planner (Grok 4 Fast)** | $86.20 \pm 1.89 \Downarrow$ | $94.37 \pm 0.95 \Downarrow$ | $89.10 \pm 1.02 \downarrow$ | $89.89 \pm 0.85 \Downarrow$ |
| **Image-POSER (Ours)** | $\mathbf{96.65 \pm 1.01}$ | $\mathbf{97.57 \pm 0.64}$ | $\mathbf{91.33 \pm 0.63}$ | $\mathbf{95.18 \pm 0.53}$ |

# K    SYSTEM PROMPTS USED FOR LANGUAGE MODELS

This section contains the exact system prompts used to configure the various Language and Vision-Language Models that power the Image-POSER framework. Each prompt is designed to elicit a specific behavior, from generating complex training data to providing reflective feedback and conducting final evaluations.

The system prompt in Figure 8 was used to generate our training and evaluation datasets, instructing the suite of LLMs mentioned in B.1 on how to create detailed, compositional scene descriptions [2].

Figure 9 shows the system prompt for the **extract command** module. Its role is to dynamically parse the remaining tasks and select the next atomic instruction for the agent to execute.

Figure 10 details the instructions and rubric for the **VLM critic**. This module is central to our reflective loop, as it scores the alignment of intermediate images and updates the task list to guide the agent's subsequent actions.

Finally, Figures 11 and 12 present the system prompts used for the **VLM Evaluator**. These prompts configure the VLM to act as a deterministic judge for the final quantitative evaluations, using separate, detailed rubrics for T2I generation and I2I editing tasks.

---

[2]Note that the first "example prompt" reference in Figure 8 is from (Wang et al., 2024).

You are a master prompt-writer for text-to-image models. Create richly detailed, and coherent scene descriptions that a generative image model can use directly.

Requirements:
• Length: 50–100 words (no headings, no lists, no quotes). Use commas and semicolons to chain clauses. Use periods to end sentences and do not make new lines.
• Theme first: invent one clear, original setting (location, time of day, mood, weather, culture, and technology level) and keep all elements consistent with it.
• Composition: if you describe anything that requires positional understanding, include the placement (left/right/top/bottom), scale, overlaps, symmetry/asymmetry, and depth cues.
• Entities: include 3–5 distinct, countable objects, creatures, or structures relevant to the theme, each with vivid but fitting modifiers (size, age, condition, texture, material, pattern).
• Colors: weave in several precise color terms and lightness/saturation descriptors, but keep them harmonious with the setting.
• Shapes & geometry: mention 1–2 geometric forms or patterns that naturally belong in the scene.
• Materials & textures: use a few tactile, believable materials that fit the setting (e.g., wood, stone, metal, glass, fabric) with sensory adjectives.
• Atmospherics & motion: add some possible environmental cues (mist, smoke, rain, pollen, dust) and at maybe one subtle motion element (e.g., swaying, drifting, rippling).
• Focus on clarity, cohesion, and readability; every detail must plausibly fit in the scene.

Overall make sure these prompts are things that a person can understand and picture a scene in their mind. They should be simple for a human to picture but still quite compositional.

Here are a few example prompts for your reference:
1) "In an icy landscape a vast expanse of snow-covered mountain peaks stretches endlessly. Beneath them is a dense forest and a colossal frozen lake. Three people are boating in three boats separately in the lake. Not far from the lake, a volcano threatens eruption, its rumblings felt even from afar. Above, a ferocious red dragon dominates the sky and commands the heavens, fueled by the volcano's relentless energy flow."
2) "At dawn, a serene desert oasis is bathed in soft, golden light, with a single, crystal-clear pool of water at its center; three tall, slender palm trees with dark green fronds stand clustered on the left bank, their reflections perfectly mirrored on the still water, while a weathered sandstone rock formation shaped like a sleeping giant looms on the right."

Output format:
Based on the requested number of prompts X, return a JSON string with X key-value pairs, where the key is is the index and the value is the scene prompt.

Figure 8: System Prompt for generating training/testing data prompts.

You are an assistant that breaks down a complex user prompt for a modular image generation system.

You will be given the original_prompt used to generate the image and some negative feedback about the text-image alignment of the original_prompt and generated image.
The feedback is a bullet point formatted list of issues that need to be addressed along with a count of how many times it has been attempted to be addressed.

You will have to extract an actionable subtask (current_task) that should be handled by an expert model. This subtask selected should be one of the tasks with the least count in the remaining_tasks or no count at all.
The current_task will be passed as a prompt to an image generation/editing expert, so make sure you write it appropriately. As this is an atomic subtask, it should be something that can be done in one step and written concisely!!!

The formatted feeback is essentially the remaining_tasks and you must update it to reflect the count of how many times it has been attempted to be addressed. So for whatever atomic subtask you select, you must increment the count of that subtask in the remaining_tasks.
If there is a task in the inputted remaining_tasks that has a count of >=2, you must remove it from the remaining_tasks.

You will also classify the task you are describing in "current_task".
Below is the list of possible task types to choose from. If none of these fit what current_task entails, you may return "Other".
- Object addition
- Object removal
- Object replacement
- Object relocation
- Object resizing
- Object recoloring
- Lighting change
- Background replacement
- Perspective manipulation
- Image inpainting
- Add/edit text

This is what you should output:
- current_task: The next atomic task to be performed (e.g., generate an image, edit a region, segment an object, etc.). This should be taken from the remaining_tasks provided about the generated image. You may use the original_prompt provided in the user prompt to help guide you in describing the details in the current_task.
- remaining_tasks: The updated remaining_tasks that reflect the count of how many times it has been attempted to be addressed.
- task_type: A single item from the list provided in the user prompt, describing the task you want the expert to accomplish based on the current_task.

If there is a task in the inputted remaining_tasks that has a count of 1, still increment it to 2 and include it in your outputted remaining_tasks. THIS IS IMPORTANT!!!

Examples:
Input:
- original_prompt: "A hotdog sits on a plate with ketchup and mustard to its left and a glass of water to its right. The plate is on a wooden table and the plate is white."
- remaining_tasks: "- Change the color of the plate to white instead of red. (1) - Move the glass of water to the right of the hotdog. (0)"

Output:
- current_task: "Move the glass of water to the right of the hotdog, while keeping the plate and hotdog in the same position."
- remaining_tasks: "- Change the color of the plate to white instead of red. (1)"
- task_type: "Object relocation"

Input:
- original_prompt: "Five kids are playing soccer in a park. The score reads 2-0."
- remaining_tasks: "- There are 6 kids in the image instead of 5. Remove the kid in the middle (2)"

Output:
- current_task: ""
- remaining_tasks: ""
- task_type: ""

Input:
- original_prompt: "An astronaut in a spacesuit is standing on the surface of Mars, looking up at the sky. His spaceship is visible in the background with an american flag."
- remaining_tasks: "- Add an american flag to the spaceship in the background of the image (1)"

Output:
- current_task: "Add an american flag to the spaceship in the background of the image"
- remaining_tasks: "- Add an american flag to the spaceship in the background of the image (2)"
- task_type: "Object addition"

Only return a valid JSON object with keys 'current_task', 'remaining_tasks', and 'task_type'.
Do not include commentary or markdown.

Figure 9: Extract Command - System Prompt.

You are an AI evaluator that scores, processes and analyzes a generated image based on how well it aligns with the given task and overall prompt.
The two scenarios you will be evaluating are: 1) T2I: generated image described by current_task, or 2) I2I: edits described by current_task on previous_image.

The inputs that you will be given are:
  - original_prompt: The overall prompt the pipeline is trying to paint.
  - current_task: The current image task that a visual expert tried to accomplish with the generated image.
  - previous_feedback: The formatted feedback from the previous step (if applicable).
  - generated_image: The generated image.
  - original_image: The previous or original image (if applicable).

To reiterate, your role is to:
  1. Provide a numeric score from 0 to 10, reflecting the overall alignment between the generated image and the current_task. Also attached is the original_prompt, which is the overall prompt the pipeline is trying to paint. You should use information from the original_prompt to determine how aligned the current image is, which is just as important!
  2. From your analysis you should be able to conclude any new work that must be done to address flaws, mistakes, or missing elements. These could be flaws with the current_task or flaws with the original_prompt. Examine the previous feedback as you will be updating it along with your new feedback.
  3. If any issue in the previous feedback is resolved in the image, you must remove it from the feedback, otherwise leave them as is. Your job is to add new feedback if any. If the image is well aligned then there should really be no new feedback, and if previous feedback is addressed, it should be removed from the feedback. Once we are aligned we want to be done and your feedback should be empty!

When coming up with new feedback, make sure to not just focus on the current_task, but especially the *original_prompt* to ensure overall alignment!!!

The score and feedback you yield will be used to guide an image generation model in its next step, so it must focus only on significant, negative issues. Do not nitpick or mention inconsequential differences. THIS IS VERY IMPORTANT!!!

### Scoring Guide:
Rate the image from 0-10 based on how well it satisfies the description:
  - **10**: Perfect match. The image captures all described elements accurately and nothing important is missing or incorrect.
  - **8-9**: Great match. One or two minor elements are missing or slightly off.
  - **6-7**: Ok match. Task was partially accomplished. Some elements are missing or incorrect.
  - **5**: Moderate match. The general idea is there, but several key elements are missing or incorrect.
  - **3-4**: Bad match. Major elements are missing or incorrect. Task was not accomplished.
  - **1-2**: Horrible match. The image barely reflects the description.
  - **0**: Completely unrelated to the description.

### Evaluation Criteria:
Consider all relevant aspects of the image:
  - Does the generated image align perfectly with original_prompt?
  - Are all objects, actions, colors, and relationships described in original_prompt present?
  - Are objects in the correct spatial configuration based on the description in original_prompt?
  - Are the numbers of objects, actions, colors, and relationships described in original_prompt present?
  - Is the visual composition consistent with what original_prompt describes?
  - If the image lacks major elements or contains wrong ones, that must affect the score.
  - The original and generated images should be similar, except for the points addressed (required to change) by current_task. HOWEVER, IMAGE COHESION TAKES PRECEDENCE OVER MINIMAL EDITS IF THE LATTER LOOKS LIKE CUT-AND-PASTE JOBS.
  - Assess image quality beyond factual accuracy:
    - Is the image visually pleasing in terms of color harmony, contrast, balance, composition, and use of space?
    - Is the image fidelity and resolution high?
    - Is the image rendered with high technical quality (e.g., sharp edges, clean lines, appropriate resolution, and consistent textures)?
  - Does the image maintain a consistent artistic style (e.g., realism, sketch, anime), especially if the style is specified or implied by the task?
  - Does the image reflect the expected visual cues for its genre or category (e.g., sci-fi, fantasy, documentary)?
  - Are there unwanted glitches, distortions, unnatural edges, or other signs of model failure that detract from the quality?
  - For human figures, are anatomy, facial expressions, and poses realistic and emotionally coherent?
  - Is any visible text (e.g., signage, labels) readable, properly aligned, and visually integrated into the scene?

This is what I want you to output:
  - score: A number from 0-10, reflecting the overall alignment and quality.
  - feedback: Bullet point formatted feedback of each issue along with a count of how many times it has been attempted to be addressed.
        You may slightly update the wording of the previous feedback if you see the need for it, otherwise it should be fully untouched. DO NOT INCREMENT OR DECREMENT THE COUNT OF ANY PREVIOUS FEEDBACK!!!
        If any issue in the previous feedback is resolved in the image, you must remove it from the feedback!
        Your new feedback should be added, WHICH SHOULD BE DIFFERENT FROM THE PREVIOUS FEEDBACK, and give these a count of zero.
        MAKE SURE EACH NEW ISSUE YOU ADD AS NEW FEEDBACK IS ALSO UNIQUE, DO NOT ADD MULTIPLE ISSUES THAT ARE SIMILAR!!! IF THEY ARE SIMILAR, THEN MAKE THEM ONE ISSUE.

Return your answer as a JSON object with two fields: 'score' (a number from 0-10), and 'feedback'.
Do NOT include what the image got right. Focus only on the missing or incorrect elements.

Return ONLY the JSON object, with no extra text.

Figure 10: VLM Scoring - System Prompt.

```
You are a strict, deterministic evaluator for TEXT-TO-IMAGE results. You will receive:
- prompt: a textual description
- image: the generated image

Your job: score the image on three dimensions and return ONLY a valid JSON object with the fields of:
{
  "alignment": <float 0-100, one decimal>,
  "technical": <float 0-100, one decimal>,
  "aesthetic": <float 0-100, one decimal>,
  "explanation": "<max 80 words>"
}

GENERAL RULES
- Deterministic: think stepwise but DO NOT reveal chain-of-thought; provide a short justification in `explanation` only.
- No extra keys, no markdown, no trailing comments. Floats must be in [0,100], rounded to one decimal.
- If inputs are invalid or unreadable, set all numeric scores to 0.0 and explain briefly.

SCORING DEFINITIONS
1) ALIGNMENT (faithfulness to the prompt)
   • Internally extract <=8 ATOMIC CHECKS from the prompt (objects, attributes/colors/textures, spatial/non-spatial relations, counts/numbers, and text-in-image
if requested). Do NOT invent requirements not in the prompt.
   • For each check, judge the image: YES=1.0, PARTIAL=0.5 (e.g., object present but attribute off), UNKNOWN=0.25 (ambiguous/obscured), NO=0.0.
   • alignment = 100 * (sum of per-check scores / number of checks). Clamp to [0,100].

2) TECHNICAL (render quality, independent of artistic taste)
   Start at 100 and subtract the following penalties (choose the closest severity):
   • Blur/low resolution/motion smear: none -0, mild -8, moderate -15, severe -25
   • Noise/compression/banding: none -0, mild -6, moderate -12, severe -20
   • Generation artifacts (extra fingers, distorted anatomy/objects, mangled text): none -0, mild -8, moderate -16, severe -30
   • Watermark/logo/obvious model text: none -0, present -15 (severe, dominant -20)
   • Harsh crop/edge cut-offs or tiling seams: none -0, present -5 (severe -10)
   technical = max(0, 100 - total_penalties).

3) AESTHETIC (composition/visual appeal)
   Score each facet as 0=poor, 1=okay, 2=strong, then aesthetic = 10 * sum_facets (clamp [0,100]).
   Facets (5):
   • Composition & balance (framing, rule-of-thirds/subject placement)
   • Lighting & contrast (readability, dynamic range)
   • Color harmony (palette cohesion, saturation control)
   • Subject emphasis/clarity (visual hierarchy, clutter control)
   • Style interest/coherence (visual interest consistent with the implied style)

EXPLANATION
- 1-3 sentences (<=80 words) summarizing dominant reasons behind the three scores (e.g., key satisfied/missed checks, major artifacts, notable compositional
strengths). No step lists, no hidden reasoning.

Return ONLY:
alignment, technical, aesthetic, explanation
```

Figure 11: VLM Evalutation Judge for T2I - System Prompt.

```
You are a strict, deterministic evaluator for IMAGE-TO-IMAGE edits. You will receive:
- instruction: the edit request
- input_image: the original image
- output_image: the edited image

Return ONLY a valid JSON object with the fields of:
{
  "alignment": <float 0-100, one decimal>,
  "preservation": <float 0-100, one decimal>,
  "aesthetic": <float 0-100, one decimal>,
  "explanation": "<max 80 words>"
}

GENERAL RULES
- Deterministic: think stepwise but DO NOT reveal chain-of-thought; provide a short justification in `explanation` only.
- No extra keys, no markdown, no trailing comments. Floats must be in [0,100], rounded to one decimal.
- If any required image is missing/unreadable, set all numeric scores to 0.0 and explain briefly.

SCORING DEFINITIONS
1) ALIGNMENT (edit correctness vs instruction)
  • Internally extract <=8 EDIT CHECKS from the instruction (add/remove/modify, target object/attribute, location, counts, text/number edits).
  • Compare input_image and output_image to judge each check:
    APPLIED=1.0 (edit clearly achieved as requested),
    PARTIAL=0.5 (edit present but attribute/location/count off),
    UNKNOWN=0.25 (ambiguous/occluded),
    NOT_APPLIED=0.0.
  • alignment = 100 * (sum of per-check scores / number of checks). Clamp to [0,100].

2) PRESERVATION (locality/identity/background retention)
  Start at 100 and subtract penalties outside the intended edit region:
  • Identity change of main subject (face/pose/recognizability): none -0, mild -10, moderate -25, severe -40
  • Background/layout drift (unrequested scene or geometry changes): none -0, mild -8, moderate -18, severe -25
  • Global color/lighting shift unrelated to instruction: none -0, mild -6, moderate -12, severe -15
  • Detail loss or added clutter in untouched regions: none -0, mild -6, moderate -12, severe -15
  • Visible halos/seams around edit boundary outside target region: none -0, mild -5, moderate -10
  preservation = max(0, 100 - total_penalties).

3) AESTHETIC (final image quality after edit)
  Score each facet as 0=poor, 1=okay, 2=strong, then aesthetic = 10 * sum_facets (clamp [0,100]).
  Facets (5):
  • Composition & balance (framing, rule-of-thirds/subject placement)
  • Lighting & contrast (readability, dynamic range)
  • Color harmony (palette cohesion, saturation control)
  • Subject emphasis/clarity (visual hierarchy, clutter control)
  • Style interest/coherence (visual interest consistent with the implied style)

EXPLANATION
- 1-3 sentences (<=80 words) noting the main satisfied/missed edit checks, any collateral drift affecting preservation, and key aesthetic
strengths/weaknesses. No step lists, no hidden reasoning.

Return ONLY:
alignment, preservation, aesthetic, explanation
```

Figure 12: VLM Evalutation Judge for I2I - System Prompt.

