# OpenReview forum: "Image-POSER: Reflective RL for Multi-Expert Image Generation and Editing"
_ICLR.cc/2026/Conference — Submitted to ICLR 2026_

### Official Review · Reviewer_cka1 · 2025-10-27

**Soundness:** 2
**Presentation:** 3
**Contribution:** 2
**Rating:** 4
**Confidence:** 4

**Summary:**

This work proposes Image-POSER, an RL framework that orchestrates multiple pretrained visual experts through a reflective decision process. The system decomposes long prompts into sub-tasks and dynamically selects the most suitable generator. The method demonstrates state-of-the-art performance on complex T2I/I2I benchmarks.

**Strengths:**

1. The method achieves strong quantitative and user-study results.
2. It proposes an insightful analysis of expert complementarity.
3. It proposes a formulation of sequential editing.

**Weaknesses:**

1. GPT-o3 is both a rewarder and an evaluator, which may bring self-evaluation bias.
2. It reports high latency. What is the underlying rationale for this computation, and which specific procedures are encompassed within each step?
3. Limited evaluation metrics. What about GenEval, DPG-Bench, MM-RewardBench?
4. The user-study sample is small (30 prompts and 14 participants). Besides, more details about the user study should be introduced.
5. No ablation on critic or command-extraction modules.
6. Does the system incorporate a reflection mechanism to address potential inaccuracies from API-based experts or even fundamental network errors?

**Questions:**

Please refer to Weaknesses.

---

> ### Author Response · Authors · 2025-11-28
> **Rebuttal By Authors (Part 1)**
>
> We thank the reviewer for the careful reading and the constructive questions.
> Below we respond point-by-point (tags **W1–W6** for weaknesses).
>
> ### Self-Evaluation Bias and Reflection Robustness (W1 / W5)
>
> We appreciate the concern regarding the potential for self-preference bias given our use of GPT-o3 as both the internal critic and external evaluator. To address this (W1) and to demonstrate the robustness of our architecture (W5), we have conducted comprehensive ablation studies detailed in the new Appendix G and Appendix H.
>
> 1. Addressing Self-Evaluation Bias (Appendix H): To rule out the possibility that our results were an artifact of GPT-o3 favoring its own outputs, we re-evaluated Image-POSER and all baselines using two independent, frontier-class VLM judges: Claude 4.5 Sonnet and Grok 4 Fast. As shown in Table 8, Image-POSER maintains a statistically significant lead in Alignment and Technical fidelity across all evaluators. While the absolute scoring distributions varied slightly between judges, the relative ranking remained consistent, confirming that our performance gains are objective and not due to evaluator bias.
> 2. Addressing Reflection Modules (Appendix G): To prove that our framework is not dependent on a specific model family, we performed an ablation study on the internal Critic and Extract Command modules. We systematically replaced GPT-o3 in these components with Claude 4.5 Sonnet and Grok 4 Fast. Results in Table 7 show that Image-POSER retains state-of-the-art performance (e.g., Alignment scores of ~96.4 with Claude vs. 96.6 with GPT-o3) regardless of the underlying model.
>
> We hope these experiments demonstrate that Image-POSER’s efficacy stems from the reflective RL orchestration policy itself, rather than reliance on, or bias from, the GPT-o3 architecture.
>
>
> ### Latency Rationale and Procedural Breakdown (W2)
>
> We acknowledge the higher latency compared to single-shot models. This is a deliberate trade-off where we exchange inference speed for autonomous error correction. Below we detail the procedures within a single time-step ($t$) and the rationale for this computational cost.
>
> Procedural Breakdown of a Step: As analyzed in Section 5.2, the majority of the latency stems from the reflective modules, not the RL agent itself. A single step consists of
> - Planning (High Latency $\approx$ 29.5s): The VLM Critic evaluates the previous image to calculate rewards and update the task queue, while the Extract Command module chooses the next atomic instruction. These involve API calls to large foundation models (GPT-o3).
> - Routing (Negligible Latency): The DQN agent selects an action. This is a lightweight MLP inference ($< 10$ms).
> - Execution (Most Latency): The selected expert (e.g., FLUX or DALL-E) generates the image (avg 62.8s).
>
> The rationale for this computation is to automate the feedback loop that currently consumes human time. On complex compositional prompts (e.g., Table 2/3), single-shot models usually fail, requiring a user to manually critique the result, re-engineer the prompt, or use inpainting tools, often taking minutes of active human effort to achieve a correct result.
>
> Our system absorbs this "human-in-the-loop" latency into an "agent-in-the-loop" process. While the wall-clock time is higher, the user's active time is zero. In practice, we would far prefer the agent to spend two minutes autonomously refining an output than to require the user to spend that or more on manual trial-and-error, particularly in professional settings where human time is substantially more valuable than agent time.

---

> > ### Author Response · Authors · 2025-11-28
> > **Rebuttal By Authors (Part 2)**
> >
> > ### Additional Benchmarking (W3)
> >
> > Thank you for the suggestions! We have carefully examined the requested benchmarks:
> >
> > 1) GenEval: We found that GenEval's prompts are short, atomic, and generally lack the multi-step compositional structure that motivates Image-POSER.
> > 2) DPG-Bench: DPG-Bench seems directly relevant, as it features dense and multi-object prompts that test precisely the compositional capabilities Image-POSER is designed to handle.
> > 3) MM-RewardBench: MM-RewardBench is designed to evaluate reward models for VLMs. Since Image-POSER does not rely on a trainable reward model for VLMs, and our VLM critic is used only for structured feedback rather than preference ranking, MM-RewardBench does not seem meaningfully aligned with our setting.
> >
> > We have added new experiments on DPG-Bench (Table 3) comparing Image-POSER to the strongest experts available in our registry: GPT-Image-1 and Gemini 2.5 Flash. We focus on these two baselines because they consistently outperformed all other baselines, thus providing the most meaningful comparison for assessing whether reflective orchestration yields improvements beyond what the best single-shot models can achieve. We found such a pragmatic choice to be necessary given the long and expensive computation cost of these experiments, as DPG-Bench consists of 1,065 prompts.
> >
> >
> > ### User Study (W4)
> >
> > We have increased our user study size to include 40 participants. Further details about the user study have been added in Section 4.4 and Appendix E, and Table 4 has been updated with new results, reflecting responses from the 40 participants, alongside tests of statistical significance.
> >
> >
> > ### API Handling (W6)
> >
> > Yes. Beyond the semantic reflection performed by the VLM critic, we implemented a robust 'catch-and-retry' mechanism for the underlying infrastructure. All API interactions (for both experts and the VLM/LLM modules) are wrapped in error handling that captures transient failures such as network timeouts, rate limits, context length overflows, and malformed outputs. We utilize an exponential backoff strategy with up to 5 retries per call, ensuring that the orchestration policy is not penalized for external API instability. We also enforce strict validation via Pydantic type checking against predefined schemas. Any malformed or incomplete responses are rejected and re-queried, instead of being propagated through the pipeline.

---

### Official Review · Reviewer_X2RP · 2025-10-31

**Soundness:** 3
**Presentation:** 3
**Contribution:** 3
**Rating:** 4
**Confidence:** 3

**Summary:**

The paper introduces a well-motivated framework that unifies multi-expert orchestration, reflection, and reinforcement learning for image generation. By viewing the coordination of expert models as a reflective RL problem, it provides a clear conceptual advancement beyond simple ensemble or prompt-routing schemes. The proposed Image-POSER system consistently outperforms both open-source and closed commercial models on compositional and instruction-following tasks, according to CLIP/BLIP metrics, VLM-based assessments, and human preference studies.

**Strengths:**

* The method trains only a lightweight DQN controller and keeps all experts fixed, requiring no retraining of large vision–language models. This modular structure makes it cost-efficient, flexible, and potentially useful as a plug-in layer for real-world multimodal systems.

* The proposed Image-POSER system consistently outperforms both open-source and closed commercial models (e.g., GPT-Image-1, Gemini) on compositional and instruction-following tasks.

**Weaknesses:**

* The reliance on GPT-o3 as both the critic in training and the automatic evaluator introduces potential bias and circularity. Since the same model family judges the system it helps train, it is unclear whether improvements reflect genuine quality gains or alignment with that evaluator’s preferences.

* The experiments are conducted on a relatively small computational scale, using short RL episodes and a limited replay buffer on a single T4 GPU. This raises concerns about policy stability, generalization, and whether the reported gains depend on specific prompt distributions or stochastic behavior of API-based experts.

* Human evaluations involve only 14 participants and focus narrowly on technical alignment and aesthetic fidelity. Broader aspects such as diversity, creativity, latency, and cost-efficiency are not deeply analyzed. Without a comprehensive user study or robustness tests, claims about real-world deployment remain tentative.

**Questions:**

N/A

---

> ### Author Response · Authors · 2025-11-28
> **Rebuttal By Authors**
>
> We thank the reviewer for the careful reading and the constructive questions.
> Below we respond point-by-point (tags **W1–W3** for weaknesses).
>
> ### VLM Choice and Evaluator Bias (W1)
>
> We agree that verifying robustness across different foundation models is essential. To address this, we have added two comprehensive ablation studies in the Appendix.
>
> **Robustness of VLM Choice:** In Appendix H (Table 7), we analyze the impact of changing the VLM/LLM used within the Image-POSER pipeline. We replaced the default GPT-o3 Critic and Extract Command modules with Claude 4.5 Sonnet and Grok 4 Fast.
> We see that Image-POSER maintains state-of-the-art performance (Average Score $>94.0$) across all configurations. Thus, the framework’s efficacy stems from the reflective orchestration policy rather than reliance on a specific model family.
>
> **Ruling Out Evaluator Bias:** To ensure our results are not an artifact of overfitting to the GPT-based critic (self-preference bias), we conducted a strict cross-evaluation in Appendix I (Table 8). We re-evaluated all methods using independent judges (Claude 4.5 Sonnet and Grok 4 Fast) completely separate from the training loop. Even under independent evaluation, Image-POSER maintains a statistically significant lead over all baselines in Alignment and Technical Fidelity. Thus, the performance gains are genuine improvements in compositional adherence, not artifacts of the evaluation metric.
>
>
> ### Concerns about Stability and Generalization (W2)
> We acknowledge the lightweight setup, but we argue that the scale is appropriate for the dimensionality of the specific learning problem we are solving.
>
> **Policy Stability & Computational Scale:** While image generation is computationally heavy, the orchestration policy operates in a low-dimensional state space. The DQN inputs are text embeddings (dim=1536), and the action space is discrete and small ($|\mathcal{A}| \approx 12$). Training a 3-layer MLP on this low-dimensional manifold is numerically stable and sample-efficient, requiring orders of magnitude less compute than training the expert models themselves. The T4 GPU is sufficient because it is only updating the lightweight router, while the heavy visual reasoning is offloaded to the frozen, pre-trained VLM.
>
> **Generalization:** Our results on held-out benchmarks (T2I-CompBench in Table 1, Custom T2I/I2I in Table 2, DPG-Bench in Table 3) demonstrate that the agent learns generalized routing strategies rather than overfitting to specific prompt content. For example, the agent learns the meta-strategy that "spatial errors are best resolved by expert X" or "style mismatches require the expert Y" regardless of whether the image contains a cat or a car. General examples of this can be seen in Appendix D Figure 7. This strategic logic transfers robustly to unseen prompts.
>
> **Stochasticity:** We account for expert stochasticity through the retry mechanism and the probabilistic nature of the Q-learning update. If an expert is unreliable, the agent learns to discount its Q-value over time. As noted in Appendix C Figure 5, the loss and reward curves converge smoothly, indicating that the agent successfully learns a robust policy despite the noise inherent in API-based experts.
>
> ### User Study and Broader Aspects (W3)
>
> We have increased our user study size to include 40 participants. Further details about the user study have been added in Section 4.4 and Appendix E, and Table 4 has been updated with new results, reflecting responses from the 40 participants, alongside tests of statistical significance.
>
> We note that alignment on long, compositional prompts is the central capability our system targets. Diversity and creativity are desirable qualities, but ultimately bounded by the pretrained expert models themselves. Since Image-POSER orchestrates rather than trains new generative models, we focus our evaluation on the dimension where orchestration can meaningfully improve over individual experts.
>
> To further validate Image-POSER's robustness beyond our initial experiments (Tables 1, 2, and 4), we introduce Table 3. This evaluation utilizes DPG-Bench, a diverse external benchmark comprising 1,065 dense prompts. Image-POSER outperforms all baselines on this dataset, confirming that the agent learns generalizable routing strategies rather than overfitting to specific training prompt distributions.

---

### Official Review · Reviewer_fcCe · 2025-10-31

**Soundness:** 2
**Presentation:** 3
**Contribution:** 2
**Rating:** 4
**Confidence:** 3

**Summary:**

The paper proposes a system that casts multi-step image generation and editing as an RL problem. An agent selects among the pretrained expert models to execute sub-steps of a complex prompt. A vision-language model critic provides dense rewards and reflective feedback, guiding the system to decompose the original prompt into successive instructions. According to the authors, Image-POSER outperforms several baselines on compositional benchmarks and human evaluations.

**Strengths:**

1. The paper addresses an important challenge in text-to-image generation: following long, compositional prompts with sequential edits. Casting this as an RL task that orchestrates multiple specialized models is an interesting angle.

2. The approach leverages existing high-quality models (e.g., diffusion models, editors) without retraining them. As noted by the authors, the only learnable part is a lightweight DQN with a 3-layer MLP, making the system relatively plug-and-play.

3. According to the reported metrics, Image-POSER achieves higher alignment and attribute-binding scores than all compared baselines. In the user study, annotators preferred Image-POSER’s outputs over baselines in the majority of cases. These results suggest the approach has merit.

**Weaknesses:**

1. Limited analysis of VLM choice. The paper lacks experiments on selecting the VLM as a judge or a critic.

2. Missing efficiency analysis. The method involves multiple iterative steps, which may be less efficient than alternatives. Please provide a clear efficiency study compared with baselines under matched settings, and analyze scaling.

3. The DQN’s state is just the current instruction and the remaining task list embedded as text, which means the agent has very limited information. In essence, the DQN is simply learning to pick the most generally capable expert for each kind of instruction, rather than truly reasoning.

4. The custom prompts for training and evaluation were all generated or guided by large LMs. This synthetic data may not reflect real user queries. The evaluation metrics also depend on the same GPT-based critic, risking overfitting to the critic’s biases.

5. The human study is quite small (14 annotators over 30 prompts) and reports only win rates; statistical significance of preferences is not discussed.

**Questions:**

1. The DQN formulation seems quite simple (3-layer MLP, 12 discrete actions). Why did you choose a DQN over simpler non-learning strategies (e.g., heuristic or LLM-based planners)? Do you have ablations showing that RL actually improves performance?

2. The paper treats image synthesis as a sequential MDP with reflective feedback. Could you formalize the state transition and reward function more rigorously? How do you ensure that the Markov assumption holds when the environment is partly determined by opaque pretrained models?

3. You limit each episode to 6 steps and drop commands after 3 failed attempts. How were these thresholds chosen? Did you experiment with different maximum step lengths or retry counts, and what was their impact on convergence or quality?

**Details Of Ethics Concerns:**

The paper has an ethical statement in the supplementary materials.

---

> ### Author Response · Authors · 2025-11-28
> **Rebuttal by Authors (Part 1)**
>
> We thank the reviewer for the careful reading and the constructive questions.
> Below we respond point-by-point (tags **W1–W5** for weaknesses, **Q1–Q3** for explicit questions).
>
> ### VLM Choice and Evaluator Bias (W1/W4)
>
> We agree that verifying robustness across different foundation models is essential. To address this, we have added two comprehensive ablation studies in the Appendix.
>
> **Robustness of VLM Choice (W1):** In Appendix H (Table 7), we analyze the impact of changing the VLM/LLM used within the Image-POSER pipeline. We replaced the default GPT-o3 Critic and Extract Command modules with Claude 4.5 Sonnet and Grok 4 Fast.
> We see that Image-POSER maintains state-of-the-art performance (Average Score $>94.0$) across all configurations. Thus, the framework’s efficacy stems from the reflective orchestration policy rather than reliance on a specific model family.
>
> **Ruling Out Evaluator Bias (W4):** To ensure our results are not an artifact of overfitting to the GPT-based critic (self-preference bias), we conducted a strict cross-evaluation in Appendix I (Table 8). We re-evaluated all methods using independent judges (Claude 4.5 Sonnet and Grok 4 Fast) completely separate from the training loop. Even under independent evaluation, Image-POSER maintains a statistically significant lead over all baselines in Alignment and Technical Fidelity. Thus, the performance gains are genuine improvements in compositional adherence, not artifacts of the evaluation metric.
>
> ### Efficiency Analysis (W2)
>
> We acknowledge that the iterative nature of Image-POSER introduces latency, but we argue that "Time to Success" is the more relevant metric for complex workflows than single-shot inference speed.
>
> As explained in Section 5.2, reflective steps introduce non-trivial latency. In our setup, the combined extract command and reward calls averaged 29.54s per step (excluding expert runtime). Regarding our average number of steps, we take 2.12 steps per episode for inference. This results in a noticeable end-to-end overhead compared to the average runtime of 62.80s for single-shot baselines.
>
> While baselines are faster per inference, they consistently fail on complex prompts (as shown in Tables 2/3/4). In real-world workflows, this failure necessitates human intervention: analyzing the error, re-engineering the prompt, and manually retrying multiple times. The overhead in Image-POSER represents the automation of this "human-in-the-loop" effort. While the wall-clock time is higher, the user's active time is zero. In practice, we would far prefer the agent to spend two minutes autonomously refining an output than to require the user to spend that or more on manual trial-and-error, particularly in professional settings where human time is substantially more valuable than agent time.
>
> We note that $T_{\max}$ is a tunable hyperparameter. Users prioritizing speed over autonomous refinement can lower this budget (e.g., $T_{\max}=2$) to reduce runtime while still benefiting from the initial orchestration choice. In the extreme case where the DQN only has budget to choose one expert, results will be as bad as the best single-shot expert available in the registry. For reference, the distribution over the number of steps taken during inference can be found in Appendix A.2 Figure 4.
>
> Alternatively, users who desire a tradeoff between time and cost can add corresponding regularization terms to the DQN's reward function, prioritizing fast models for time-sensitive users or free models for cost-sensitive users. This follows from the same pattern we used to add $-0.05t$, a regularization term for longer pipelines, to the DQN's reward function (Equation 3).
>
>
> ### DQN State (W3)
>
> We acknowledge that the text-only state is a compressed representation and does not retain pixel-level information (i.e., it is a Partially Observable MDP). However, we argue that for the specific task of expert orchestration, semantic state is more bandwidth-efficient than raw visual state.
>
> The VLM Critic acts as a "semantic sensor," translating complex visual states ($I_t$) and defects into explicit textual constraints ($c_t$). The hypothesis of Image-POSER is that the decision to switch experts depends less on pixel statistics and more on semantic failure modes (e.g., "the object is distorted," "the style is incorrect," or "the count is wrong").
>
> By relying on the VLM to extract these diagnostic features into the textual state, the DQN is relieved of the burden of learning computer vision from scratch. Instead, it learns a high-level routing policy: mapping specific types of semantic failures (encoded in the text) to the experts best suited to resolve them. This abstraction allows the agent to be lightweight and sample-efficient while leveraging the massive visual reasoning capabilities of the VLM.
>
> We have reflected this in Section 3.2 of the revised paper.

---

> > ### Author Response · Authors · 2025-11-28
> > **Rebuttal by Authors (Part 2)**
> >
> > ### Validity of Data & Prompt Distribution (W4)
> >
> > We acknowledge the concern regarding synthetic data and have taken specific measures to ensure our evaluation reflects diverse, realistic usage:
> >
> > 1. Diversity & Human Curation (Appendix B.1): Our T2I dataset was not generated by a single model but sampled from an ensemble of 7 distinct LLMs (including GPT-5, Claude 3, and Llama 4) to maximize stylistic diversity, manual filtering. Furthermore, our I2I benchmark is grounded entirely in real images (from HQ-Edit and ImgEdit) with manually authored instructions, ensuring the model handles realistic visual distributions.
> > 2. Generalization to Public Benchmarks: To verify that our policy does not overfit to our internal data, we evaluated Image-POSER on DPG-Bench (Section 4 Table 3), an external, standard benchmark for dense compositional reasoning. Our state-of-the-art performance here confirms robustness on established tasks.
> > 3. Human Alignment: Crucially, our user study (Section 4.4 Table 4) shows that human participants consistently prefer Image-POSER over baselines. The strong alignment between human preference and our automated metrics indicates that our results reflect genuine perceptual quality rather than artifacts of synthetic prompt engineering.
> >
> >
> > ### User Study (W5)
> >
> > We have increased our user study size to include 40 participants. Further details about the user study have been added in Section 4.4 and Appendix E, and Table 4 has been updated with new results, reflecting responses from the 40 participants, alongside tests of statistical significance.
> >
> >
> > ### Ablation with LLM Planner (Q1)
> >
> > We opted for a reinforcement learning approach because expert orchestration is fundamentally a decision-making problem under uncertainty, where the "reliability" of an expert is stochastic and often opaque to zero-shot reasoning.
> >
> > Our design choice for the lightweight DQN itself is rooted in how state information is constructed within Image-POSER. Both the extract-command and critic modules are reasoning-capable models with full visibility into the entire Image-POSER pipeline, explicitly instructed to produce structured and faithful descriptions of all visually and semantically relevant information at each step. This keeps the DQN’s role well-scoped, only needing to learn which expert to invoke next, conditioned on a complete state representation of the current environment. In this setting, we found that a 3-layer MLP is sufficient for learning effective policies, as can be seen by the loss/reward curves in Appendix C Figure 5.
> >
> > To directly evaluate whether RL provides value over non-learning strategies, we have added an ablation (Appendix J) comparing Image-POSER against an LLM-based planner, as requested.
> >
> > Experimental Setup: We replaced the DQN agent with Grok 4 Fast, prompting it to act as a planner. This planner was given full access to the prompt and the critic's feedback. It was tasked with (i) selecting the initial T2I expert and (ii) reflecting on feedback and logically scheduling all subsequent edits by assigning tasks to experts based on its internal knowledge of their capabilities.
> >
> > Key Findings:
> > 1. Image-POSER surpasses the LLM Planner by over 5 points in average score ($p < 0.01$).
> > 2. Interestingly, the LLM Planner performed worse than the single strongest model (GPT Image 1). This indicates that the planner often "over-thinks" the pipeline, routing tasks to specialized editors that theoretically should work but practically fail on specific edge cases.
> >
> > We hope this ablation clarifies that the lightweight DQN is not merely a simplification, but a necessary component for adapting to the real-world stochasticity of heterogeneous expert models.

---

> > > ### Author Response · Authors · 2025-11-28
> > > **Rebuttal by Authors (Part 3)**
> > >
> > > ### Formalizing MDP (Q2)
> > >
> > > We appreciate the reviewer’s suggestion to rigorously formalize the environment. We have updated Section 3 (Methodology) to explicitly define the generation process as a Markov Decision Process (MDP) $\mathcal{M} = \langle \mathcal{S}, \mathcal{A}, \mathcal{T}, \mathcal{R}, \gamma \rangle$.
> > >
> > > We define the state at time $t$ as a tuple $s_t = (\mathcal{P}, I_t, c_t, C_t^{\text{rem}})$, containing the global prompt $\mathcal{P}$, the current image $I_t$, the active command $c_t$, and the set of remaining commands $C_t^{\text{rem}}$. The action $a_t$ selects an expert $e \in \mathcal{E}$.
> > >
> > > The transition function $\mathcal{T}(s_{t+1} | s_t, a_t)$ is decomposed into one deterministic and two stochastic functions corresponding to our system components:
> > >
> > > 1. Identity: The global prompt remains constant: $\mathcal{P}_{t+1} = \mathcal{P}_t$ (deterministic).
> > > 2. Visual Execution (Expert): The next image is sampled from the selected expert's distribution: $I_{t+1} \sim e_{a_t}(I_t, c_t)$ (stochastic).
> > > 3. Reflective Planning (VLM/LLM): The VLM critic and LLM command extractor update the task queue based on the new visual state: $(c_{t+1}, C_{t+1}^{\text{rem}}) \sim f_{\text{reflect}}(I_{t+1}, c_t, C_t^{\text{rem}}, \mathcal{P})$ (stochastic).
> > >
> > >
> > > On the Markov assumption, we argue that it holds because the experts (T2I/I2I models) and the VLM critics operate as stateless functions during inference. Their output depends exclusively on their immediate inputs (the current image $I_t$ and text $c_t$). They do not maintain hidden internal states or memory across time steps. Therefore, the generated state provides a sufficient statistic for the future evolution of the system, satisfying the Markov property. The "opacity" of the models introduces stochasticity into the transition function $\mathcal{T}$, which RL is well-suited to handle.
> > >
> > >
> > > ### Step Budget and Retry Limits (Q3)
> > >
> > > We chose the thresholds for $T_{\max}$ and the retry limit based on a combination of dataset characteristics, quality preservation, and resource constraints. We have added Appendix A.2 and Figure 4 (Time Step Histogram) to the paper to analyze this in detail.
> > >
> > > **Maximum Step Budget ($T_{\max}=6$):** It is important to note that $T_{\max}$ is a hard ceiling; the agent is trained to terminate the episode as soon as the VLM critic is satisfied. Our choice was driven by:
> > > 1. Prompt Complexity: Our dataset contains prompts with 3–5 distinct constraints. A budget of 6 ensures we can handle the most complex cases while allowing a margin for error correction.
> > > 2. Empirical Usage: The agent averages 2.12 steps per episode during inference and rarely approaches the limit. A lower budget (e.g., $T=3$) would have been insufficient for completion, while a higher budget was largely unutilized.
> > > 3. Quality and Cost: Qualitatively, we observed that exceeding $\sim5$ sequential I2I edits leads to "generational drift" and pixel degradation in current diffusion models. Additionally, limiting the horizon is necessary to manage the latency and monetary costs associated with the reflective VLM calls to stay within our budgets.
> > >
> > > **Retry Limit (3 Attempts):** This threshold was selected to differentiate between aleatoric uncertainty and epistemic limitation. If an expert fails once, it may be a random seed issue resolvable by a retry. However, repeated failures suggest the model fundamentally cannot perform the task or that the VLM feedback is noisy. We established 3 as a practical budget to prevent the agent from getting stuck in unresolvable loops. This is coupled with our planning logic: the Extract Command module is instructed to prioritize tasks with fewer attempts, encouraging the agent to solve easier constraints before retrying difficult ones.

---

### Official Review · Reviewer_VoMS · 2025-11-01

**Soundness:** 3
**Presentation:** 4
**Contribution:** 3
**Rating:** 10
**Confidence:** 3

**Summary:**

This paper presents Image-POSER , a reflective reinforcement learning framework designed to enhance multi-expert image generation and editing by effectively managing long-form prompts through dynamic task decomposition and expert orchestration.

The presented work addresses limitations of current text-to-image models that struggle with complex prompts. In particular it deals with long, compositional prompts that consist of multiple objects, spatial relations, or sequential edits.

An exhaustive evaluation is presented, demonstrating superior performance than baselines on spatial reasoning. The framework demonstrates the advantages of reflective orchestration in achieving high-quality outputs outperforming single generators on complex compositional tasks.

**Strengths:**

The work proposes a novel formulation of image generation and editing. It proposes a different approach than traditional ones that predefine a sequence of subtasks. The novelty of this work is to formulate image generation as a sequential decision-making problem. This formulation allows for dynamic and adaptive orchestration, with the use of a reflective reinforcement learning environment. In this way, commands are incrementally created and the outputs validated.

By orchestrating multiple pretrained models, the proposed method  leverages existing strengths of specialized systems, potentially achieving higher quality and versatility than single-model approaches.

The proposed architecture outperforms baselines (both single-model and agentic) on compositional benchmarks, indicating robust performance in complex scenarios. Additionally, user studies are reported, with superior acceptance regarding the state of the art.

**Weaknesses:**

There is a dependence on pretrained models. The orchestration relies on pretrained models. There is a risk of introducing inconsistencies, redundancies or conflicts. How the system deals with it?

A drawback of the proposed approach is its high computational cost, as the reflective steps introduce significant latency which poses challenges for practical deployment and real-time applications.

**Questions:**

The framework is described as moving toward “generalist visual agents,” but it’s unclear how well it generalizes across diverse domains, image types, or prompt complexities, beyond the tested benchmarks. It would be good to provide further discussion on this vision.

The user study is interesting. Can you provide more details on the user profiles. Are they graphical designers? Computer scientists? … Do their profile influence in the assessment they provide?

---

> ### Author Response · Authors · 2025-11-28
> **Rebuttal By Authors (Part 1)**
>
> We thank the reviewer for the careful reading and the constructive questions.
> Below we respond point-by-point (tags **W1–W2** for weaknesses, **Q1–Q2** for explicit questions).
>
> ### Reliance on Pretrained Experts (W1)
>
> The reliance on pretrained experts is a deliberate design choice to ensure modularity and plug-and-play behaviour for the user. We mitigate the risks of conflict and redundancy through three specific mechanisms:
>
> 1. Preventing Conflicts via Decomposition: The Extract Command LLM (Section 3.3) is explicitly instructed to decompose global prompts into atomic, non-overlapping sub-tasks (see System Prompt in Figure 9, Appendix K). This ensures that experts receive focused instructions rather than conflicting, multi-objective prompts.
> 2. Correcting Inconsistencies via Reflection: The VLM Critic (Algorithm 1) evaluates every generated step ($I_{t+1}$) against both the current instruction and the global prompt. If an expert introduces an inconsistency, the critic assigns a low reward (see Section 3.3), teaching the DQN agent to avoid that expert for similar states in the future.
> 3. Eliminating Redundancy via Regularization: To prevent the agent from taking unnecessary steps or using redundant experts, we include a time-step penalty term ($-0.05t$) in the reward function (Section 3.3 Equation 3). This effectively regularizes the policy, forcing the agent to learn the most concise orchestration path.
>
>
> ### Latency Analysis (W2)
>
> We acknowledge that the iterative nature of Image-POSER introduces latency, but we argue that "Time to Success" is the more relevant metric for complex workflows than single-shot inference speed.
>
> As explained in Section 5.2, reflective steps introduce non-trivial latency. In our setup, the combined extract command and reward calls averaged 29.54s per step (excluding expert runtime). Regarding our average number of steps, we take 2.12 steps per episode for inference. This results in a noticeable end-to-end overhead compared to the average runtime of 62.80s for single-shot baselines.
>
> While baselines are faster per inference, they consistently fail on complex prompts (as shown in Tables 2/3/4). In real-world workflows, this failure necessitates human intervention: analyzing the error, re-engineering the prompt, and manually retrying multiple times. The overhead in Image-POSER represents the automation of this "human-in-the-loop" effort. While the wall-clock time is higher, the user's active time is zero. In practice, we would far prefer the agent to spend two minutes autonomously refining an output than to require the user to spend that or more on manual trial-and-error, particularly in professional settings where human time is substantially more valuable than agent time.
>
> We note that $T_{\max}$ is a tunable hyperparameter. Users prioritizing speed over autonomous refinement can lower this budget (e.g., $T_{\max}=2$) to reduce runtime while still benefiting from the initial orchestration choice. In the extreme case where the DQN only has budget to choose one expert, results will be as bad as the best single-shot expert available in the registry. For reference, the distribution over the number of steps taken during inference can be found in Appendix A.2 Figure 4.
>
> Alternatively, users who desire a tradeoff between time and cost can add corresponding regularization terms to the DQN's reward function, prioritizing fast models for time-sensitive users or free models for cost-sensitive users. This follows from the same pattern we used to add $-0.05t$, a regularization term for longer pipelines, to the DQN's reward function (Equation 3).

---

> ### Author Response · Authors · 2025-11-28
> **Rebuttal By Authors (Part 2)**
>
> ### Generalization (Q1)
>
> Our vision of a "generalist visual agent" is defined not by a single model's parameters, but by our system's ability to utilize diverse tools. Our framework achieves generalization by decoupling semantic understanding (handled by frozen foundation models) from execution routing (handled by the learned policy). We see this functioning in three ways:
>
> 1. Inherited Generalization: The components responsible for interpreting diverse domains (the VLM Critic and Extract Command LLM) and generating visuals (the Expert Registry) are frozen, pre-trained frontier models. Image-POSER implicitly inherits their generalization capabilities. Whether the prompt requires "cartoon style" or "photorealism," the VLM understands it and the experts generate it without requiring our framework to be fine-tuned on those specific domains.
> 2. Domain-Agnostic Routing: The only learned component is the DQN, but importantly, it does not learn semantic content (e.g., "how to draw a car"). Instead, it maps abstract task types to experts (e.g., "Object Insertion" $\to$ Expert A, "Style Transfer" $\to$ Expert B). As noted in our analysis (Appendix D, Figure 7), the DQN learns to exploit patterns in expert capabilities (e.g., FLUX for lighting, GPT-Image-1 for textual edits). Furthermore, since the Extract Command module strips away prompt-specific "fluff" to present atomic instructions, the policy generalizes to unseen prompts as long as they decompose into familiar operations (e.g. insert, remove, recolor).
> 3. Empirical Verification: To ensure this theoretical generalization holds in practice, we moved beyond our training distribution by evaluating on DPG-Bench (Section 4.2 Table 3), a diverse external benchmark containing 1,065 dense prompts. Our state-of-the-art performance there confirms that the system generalizes well to high-complexity inputs outside our custom dataset.
>
>
> ### User Study (Q2)
>
> Thank you for your interest in our user study! We have since increased its size to include 40 participants, and further details about it have been added in Section 4.4 and Appendix E. Table 4 has also been updated with new results, reflecting responses from the 40 participants, alongside tests of statistical significance.
>
> In our analysis, we did not notice any patterns between the participants' profiles and their ratings. This could be because, as outlined in Appendix E, 1) none of the participants were experts in media generation, 2) the sample size required to test out this hypothesis is much larger than what we used, or 3) because the user profiles likely do not have an impact on their ratings. Regardless, this is not a focus of our paper, so we didn't allocate resources to explicitly prove/disprove the hypothesis.

---

### Meta-Review · Area_Chair_4j6m · 2026-01-09

**Summary:**

This paper presents Image-POSER, an RL-based framework for multi-expert image generation and editing that handles long-form prompts via task decomposition and expert orchestration. The paper received divergent scores (4, 4, 4, and 10). Reviewers noted strengths including a novel problem formulation and improved performance over single-model and agentic baselines on compositional benchmarks, supported by user studies. However, some concerns may still remain regarding DQN's state design and small-scale experiments. After considering the rebuttal, the AC thinks that overall, the flaws slightly outweigh the merits, and recommends rejection at this time.

**Reviewer Concerns:**

Concerns adequately addressed:

1. User study scale. The authors expanded the user study to 40 participants during rebuttal and reported updated results.

2. VLM choice and evaluator bias. Multiple reviewers asked this question, and the authors provided empirical observations regarding using both GPT-o3 as reward model and evaluator, etc.

3. Latency and efficiency analysis. This has also been asked by several reviewers.


Concerns insufficiently addressed:

1. The DQN’s state is just the current instruction and the remaining task list embedded as text, which means the agent has very limited information. In essence, the DQN is simply learning to pick the most generally capable expert for each kind of instruction, rather than truly reasoning.

2. The experiments are conducted on a relatively small computational scale, using short RL episodes and a limited replay buffer on a single T4 GPU. This raises concerns about policy stability, generalization, and whether the reported gains depend on specific prompt distributions or stochastic behavior of API-based experts.

3. The AC is also interested in understanding the strength and weaknesses of each image generation and editing model, and  understanding the pattern the agent has learned on how to select different experts in dealing with different prompts. The authors provided some analysis in Appendix Figure 7, but it seems not enough.

**Reviewer Scores:**

Overall, the rebuttal addresses some practical concerns, but may not be convincing enough. The AC acknowledges that this is an interesting idea, but the flaws slightly outweigh the merits, and the authors are encouraged to refine the paper and submit to future conferences.

---

### Decision · Program_Chairs · 2026-01-26

Reject